# NUPR1 protects against hyperPARylation-dependent cell death

Patricia Santofimia-Castaño[1], Can Huang[1], Xi Liu[1], Yi Xia[2], Stephane Audebert [1], Luc Camoin [1], Ling Peng [3], Gwen Lomberk [4], Raul Urrutia [5], Philippe Soubeyran [1], Jose Luis Neira[6] & Juan Iovanna [1✉]

Proteomic, cellular and biochemical analysis of the stress protein NUPR1 reveals that it binds to PARP1 into the nucleus and inhibits PARP1 activity in vitro. Mutations on residues Ala33 or Thr68 of NUPR1 or treatment with its inhibitor ZZW-115 inhibits this effect. PARylation induced by 5-fluorouracil (5-FU) treatment is strongly enhanced by ZZW-115 and associated with a decrease of $NAD^+/NADH$ ratio and rescued by the PARP inhibitor olaparib. Cell death induced by ZZW-115 treatment of pancreas cancer-derived cells is rescued by olaparib and improved with PARG inhibitor PDD00017273. The mitochondrial catastrophe induced by ZZW-115 treatment or by genetic inactivation of NUPR1 is associated to a hyperPARylation of the mitochondria, disorganization of the mitochondrial network, mitochondrial membrane potential decrease, and with increase of superoxide production, intracellular level of reactive oxygen species (ROS) and cytosolic levels of $Ca^{2+}$. These features are rescued by olaparib or $NAD^+$ precursor nicotinamide mononucleotide in a dose-dependent manner and partially by antioxidants treatments. In conclusion, inactivation of NUPR1 induces a hyperPARylation, which in turn, induces a mitochondrial catastrophe and consequently a cell death through a non-canonical Parthanatos, since apoptosis inducing-factor (AIF) is not translocated out of the mitochondria.

[1] Centre de Recherche en Cancérologie de Marseille (CRCM), Parc Scientifique et Technologique de Luminy, INSERM U1068, CNRS UMR 7258, Institut Paoli-Calmettes, Aix-Marseille Université, Marseille, France. [2] Chongqing Key Laboratory of Natural Product Synthesis and Drug Research, School of Pharmaceutical Sciences, Chongqing University, Chongqing, PR China. [3] Aix-Marseille Université, CNRS, Centre Interdisciplinaire de Nanoscience de Marseille, UMR 7325, «Equipe Labellisée Ligue Contre le Cancer», Parc Scientifique et Technologique de Luminy, Aix-Marseille Université, Marseille, France. [4] Division of Research, Department of Surgery, Medical College of Wisconsin, Milwaukee, WI, USA. [5] Genomic Science and Precision Medicine Center (GSPMC), Medical College of Wisconsin, Milwaukee, WI, USA. [6] Instituto de Biología Molecular y Celular, Edificio Torregaitán, Universidad Miguel Hernández, Elche, Alicante, Spain. ✉email: juan.iovanna@inserm.fr

PARP1 is an abundant nuclear protein which catalyzes the covalent attachment of PAR (Poly-ADP-ribose) polymers on itself and other acceptor proteins using $NAD^+$ as a donor of ADP-ribose units[1]. The generation of PAR following stresses, such as genotoxic, oxidative, oncogenic or metabolic stresses, is an extremely rapid process. In addition, PAR is also rapidly catabolized by Poly(ADP-Ribose) Glycohydrolase (PARG) indicating that PARylation is a very dynamic process[2]. Convincing data suggest that PARP1, through its physical association with or by PARylation of partner proteins, regulates major cellular functions including chromatin structure, DNA metabolism and gene expression[3]. Extraordinarily, damaged DNA stimulates the PARP1 activity. The synthesis of PAR by PARP1 in response to diverse stress stimuli triggers a signal departing from chromatin to regulate the downstream transcriptional program and the fine tuning of DNA-repairing machinery. Mechanistically, as a negatively charged polymer, PAR alters the biochemical properties of modified or interacting proteins, modulating their structure, function, and localization[4]. However, it is important to note that although the PARylation is a pleiotropic regulator of some cellular functions, when it becomes uncontrolled it may also lead to cell death as a consequence of depletion of cellular $NAD^+$ pools via a metabolic catastrophe. $NAD^+$/NADH ratio balance is essential for many metabolic reactions, such as central carbon metabolism or nucleotides and amino acid synthesis. Importantly $NAD^+$ regeneration is a limiting point for tumoral cell proliferation[5]. This is why PARP1 activity and PARylation must be perfectly regulated.

Nuclear Protein 1 (NUPR1) is a nuclear intrinsically disordered protein (IDP) of 82 amino acids that plays an important role in stress response in several tissues[6]. NUPR1 was first described because it is activated during the acute phase of pancreatitis[7]. Then, it has been shown that NUPR1 is overexpressed in response to some, if not all stresses, including minimal stresses[8]. Finally, NUPR1 was found to be activated in several cancerous tissues in which its expression was essential for their development and progression[9].

By using a multidisciplinary approach combining biophysics, chemistry, bioinformatics and biology, we have developed a small compound, named ZZW-115[10], which binds to Ala33 and Thr68 of NUPR1, hampering its interaction with importins, and consequently inhibits the nuclear translocation and its nuclear activity[11]. Treatment of cancer cells with ZZW-115 induces a strong decrease of ATP, with a reduction in OXPHOS metabolism, a switch towards anaerobic glycolysis, overproduction of ROS cell triggering cell death partially sensitive to necrostatin-1, Z-VAD-FMK and ferrostatin-1[12]. In this study, we demonstrate that NUPR1 was bound in cellulo and in vitro to PARP1 and regulates its enzymatic activity, preventing deleterious and critical hyperPARylation. Moreover, we observed that treatment of the cells with the NUPR1 inhibitor ZZW-115 abolished the protective effect of NUPR1 as a PARP1 activity regulator, inducing strong hyperPARylation and cell death mediated by a strong mitochondrial dysfunction, which was reversed by olaparib or the $NAD^+$ precursor nicotinamide mononucleotide (NMN).

## Results

### NUPR1 binds to PARP1.
We performed immunoprecipitation for NUPR1 to determine its interactome. Flag-tagged NUPR1 or GFP-tagged NUPR1 were expressed in MiaPaCa-2 cells and 24 h later we performed the immunoprecipitation for Flag or GFP either under normal conditions or following metabolic stresses induced by 24 h of glucose starvation. NUPR1-associated proteins were then identified by mass spectrometry resulting in 656 Flag-tagged NUPR1-interacting proteins under normal conditions,

and 1152 proteins under glucose starvation (Supplementary Data 1) and 271 and 530 proteins for GFP-tagged NUPR1 under normal growth conditions and glucose starvation, respectively (Supplementary Data 2). Interaction of NUPR1 with some proteins was found to be increased under metabolic stress; among them, the interaction with PARP1 shows to be one of the most important as presented in Fig. 1a and suggesting a possible role of this interaction under stress conditions.

We then validated this interaction by transfecting GFP-tagged NUPR1 and PARP1-Flag expression plasmids in MiaPaCa-2 cells and precipitating with anti-GFP or anti-Flag and revealing with anti-Flag or anti-GFP respectively. The interaction between NUPR1 and PARP1 was confirmed showing, on one hand, the precipitation with anti-GFP pull down PARP1-Flag and, on the other hand, the precipitation with anti-Flag pull down NUPR1-GFP (Fig. 1b). Interestingly, when cells were treated with the NUPR1 inhibitor compound ZZW-115, the interaction between NUPR1 and PARP1 was decreased. This result suggested that ZZW-115 and PARP1 were competing for the same binding size in NUPR1, that is, the region involving Ala33 and Thr68 (see below).

To show this interaction took place within the cell, the association between NUPR1 and PARP1 was examined using a proximity ligation assay (PLA). Confocal fluorescent microscopy analysis revealed PLA-positive foci in the nucleus indicating that NUPR1 interacts with PARP1. Importantly, this interaction was strongly decreased when cells were treated with ZZW-115 from $59.55 \pm 11.77$ foci/cell in control condition to $12.52 \pm 4.63$ under treatment with ZZW-115 (Fig. 1c). In addition, 5-FU (5-Fluorouracil) incubation increased the number of foci of interaction up to $94.73 \pm 14.81$, whereas olaparib did not show any significant effect. However, even under 5-FU + Olaparib-treatement, NUPR1 was able to interact with PARP1. In addition, in order to better understand if NUPR1 was also bound to PARylated PARP1, a PLA using NUPR1 and PAR antibodies was performed. We observed that NUPR1 and PARylated PARP1 were capable of binding under 5-FU treatment, since it is a drug triggering DNA damage and PARP1 activity. These findings confirm that NUPR1 interacts with PARP1 and PARylated PARP1, into the nucleus, and this interaction is sensitive to ZZW-115 (Supplementary Fig. 1a). In addition, we transfected MiaPaCa-2 cells with Flag wild-type (WT-NUPR1), the single mutant proteins Thr68Gln or Ala33Gln, or the double mutant Thr68Gln/Ala33Gln constructs and performed a PLA using PARP1 and Flag primary antibodies. Our results showed that PARP1 interacts with greater affinity with WT-NUPR1 (Supplementary Fig. 1b).

We also showed that the interaction took place in vitro by using recombinant proteins. We studied the interaction by Co-IP experiments, where we validated the previous observation in cells, that NUPR1 is binding to PARP1 (Supplementary Fig. 1c). We also studied the binding of both proteins by using two spectroscopic techniques: far-UV circular dichroism (CD) and fluorescence. In each technique, we acquired the spectra of NUPR1 and PARP1 in isolation, and we further acquired that of the complex formed by the two proteins at the same concentrations used when isolated. The samples used for the far-UV CD and fluorescence experiments were the same. We observed large changes between the addition spectrum (obtained by addition of the spectra of each isolated protein) and that of the complex, suggesting that: (i) there was binding between both proteins; and (ii) there were changes in the secondary structure of, at least, one of the proteins upon binding (Fig. 1d). On the other hand, the addition fluorescence spectrum was similar to that of the complex, except for small changes at shorter wavelengths, where the tyrosine residues of NUPR1 (Tyr30 and Tyr36) and those of PARP1

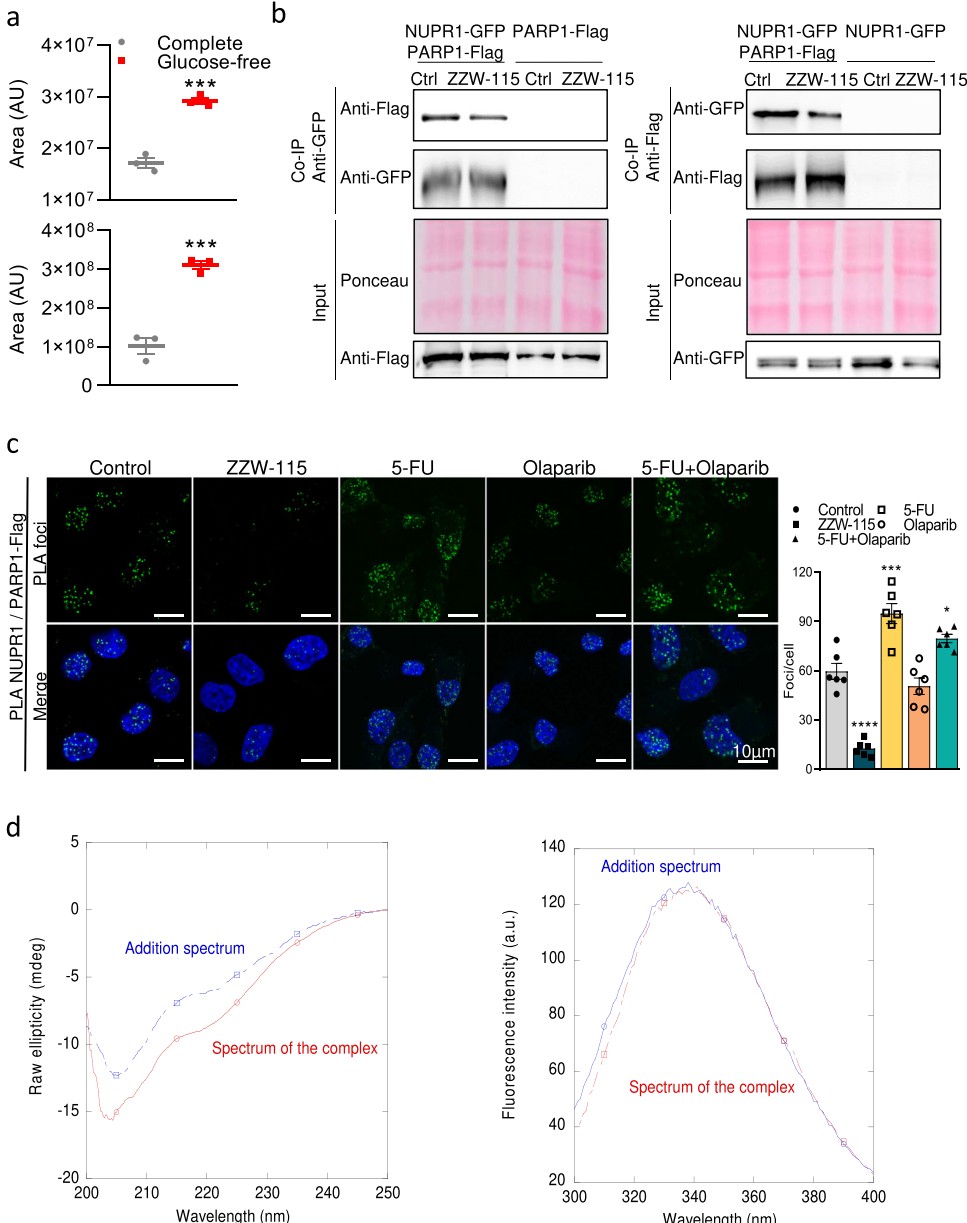

**Fig. 1 NUPR1 interacted with PARP1 in cellulo and in vitro. a** Mean of peak area values measurement after immunoprecipitation with anti-FLAG or anti-GFP agarose beads and LC-MS/MS proteomic analysis in MiaPaCa-2 cells transfected with NUPR1-Flag plasmid (up) or NUPR1-GFP (down) ($n = 3$). **b** Co-IP assays showing interactions of NUPR1-GFP with PARP1-Flag in MiaPaCa-2 cells transfected with PARP1-Flag and NUPR1-GFP plasmids. Anti-FLAG agarose beads (right) or anti-GFP agarose beads (left) were used ($n = 3$). **c** PLA was performed in MiaPaCa-2 cells in the presence or in the absence of ZZW-115 at 1.5 μM, 5-FU at 10 μM, olaparib at 25 μM or 5-FU + olaparib for 24 h. Cells were transfected with the plasmid expressing PARP1-Flag. Mouse anti-Flag and rabbit anti-NUPR1 antibodies were used. A representative experiment is shown ($n = 6$). ImageJ was used to count the number of green dots. Data represent mean ± SEM. One-way ANOVA, Dunnett correction; *$P < 0.05$; **$P < 0.01$; ***$P < 0.001$; ****$P < 0.001$. **d** Monitoring binding by spectroscopic techniques: (Left) Far-UV CD spectrum of the mixture of PARP1 and NUPR1 and that obtained by the addition of the spectra of the isolated macromolecules. (Right) Fluorescence spectrum of the same mixture of PARP1 and NUPR1 and that obtained by the addition of the spectra of the isolated macromolecules.

appear (Fig. 1d), thus confirming the involvement in the binding of region around Ala33.

**NUPR1 inhibits the PARP1 activity and ZZW-115 increased PARylation**. We hypothesized that NUPR1 might modify the PARP1 activity through its binding. To validate this hypothesis, we first supplemented with 2 and 4 μM of recombinant NUPR1 (rNUPR1) an acellular PARP1 activity assay containing increasing concentrations of PARP1. Remarkably, we found a dose-response inhibition effect of 26.5% and 48.4% with 2 and 4 μM of

rNUPR1 respectively, as presented in Fig. 2a. These results suggest a PARP1 inhibitor role of NUPR1 in vitro. In our previous work, we demonstrated that NUPR1 interact with other proteins[11,13,14] or drugs[10,15], by the regions around the Ala33 and Thr68 residues and, most importantly, mutations of these residues, hamper their interactions both in vitro and in cellulo. Thus, we used the wild-type (WT-NUPR1), the single mutant proteins Thr68Gln or Ala33Gln, or the double mutant Thr68Gln/Ala33Gln to test their effect on PARP1 activity. We also tested the effect of the ZZW-115 inhibitor. As it is shown in the Fig. 2b and

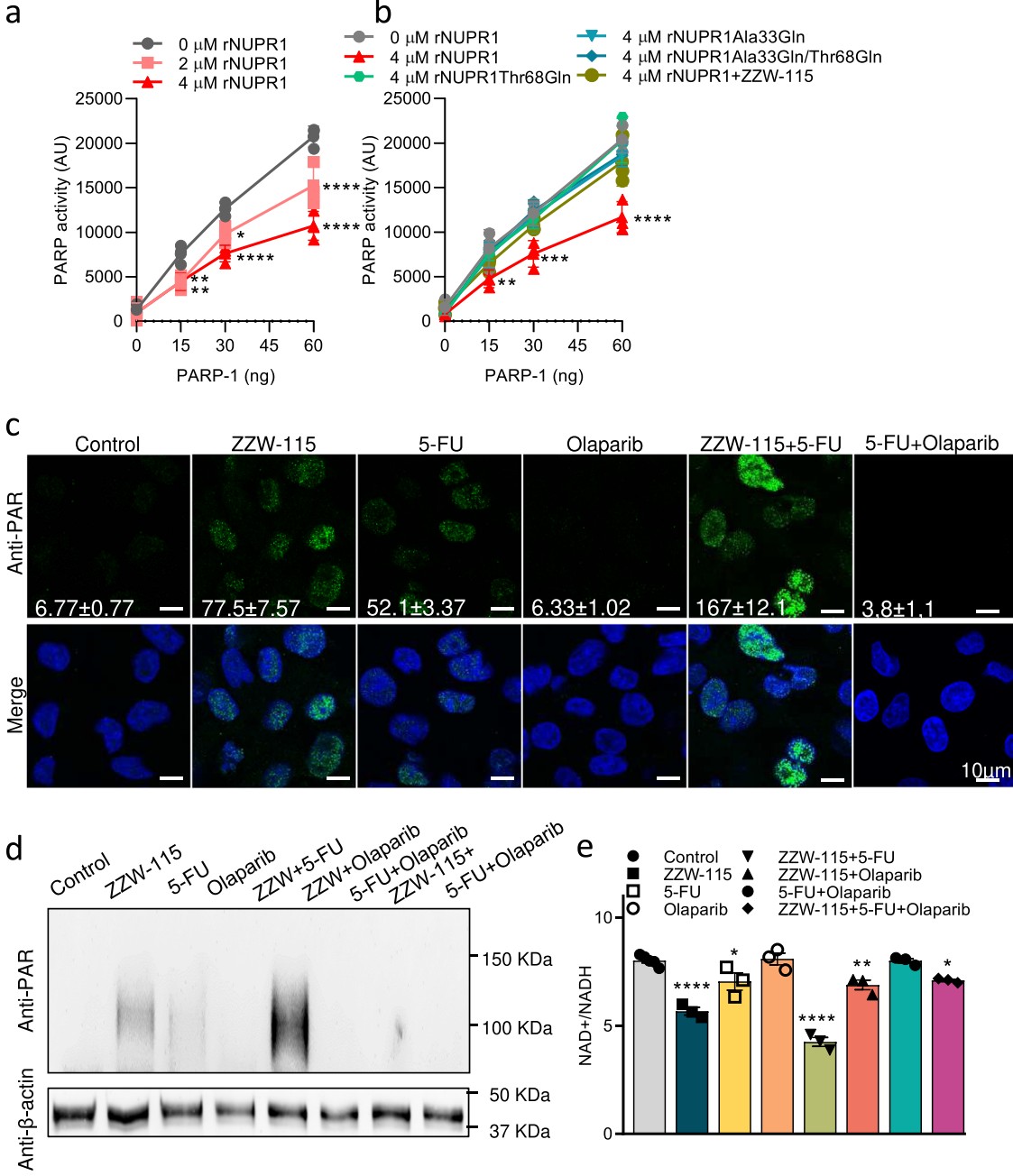

**Fig. 2 NUPR1 inhibition increased PARP1 activity in vitro and in cellulo. a** PARP1 enzymatic activity was measured in vitro alone or in combination with 2 or 4 µM of recombinant WT-NUPR1 ($n = 3$). **b** PARP1 enzymatic activity was measured in vitro alone or in combination with 4 µM of recombinant wild-type-NUPR1, or mutants recombinant NUPR1 proteins (Thr68Gln, Ala33Gln, Thr68Gln/Ala33Gln) and wild-type recombinant NUPR1 in the presence of ZZW-115 ($n = 3$). Data represent mean ± SEM. Two-way ANOVA with Sidak correction *$P < 0.05$; **$P < 0.01$; ***$P < 0.001$; ****$P < 0.001$. **c** Immunofluorescence was performed in MiaPaCa-2 cells in the presence or absence of ZZW-115 at 1.5 µM, 5-FU at 10 µM, olaparib at 25 µM, ZZW-115 together with 5-FU or 5-FU in combination with olaparib. Mouse anti-PAR and Alexa 488-labeled goat anti-mouse secondary antibodies were used. A representative experiment is shown ($n = 3$). ImageJ was used to count the number of green dots. Mean ± SEM of foci/nucleus are included. **d** Western blot analysis was performed in MiaPaCa-2 cells to evaluated PAR levels after treatments during 24 h (ZZW-115 at 1.5 µM, 5-FU) at 10 µM, olaparib at 25 µM or their combination) ($n = 3$). **e** NAD$^+$/NADH ratio was measured after 24 h treatment at the concentrations previously described ($n = 3$). Data represent mean ± SEM. One-way ANOVA, Dunnett correction; *$P < 0.05$; **$P < 0.01$; ***$P < 0.001$; ****$P < 0.001$.

Supplementary Fig. 2, mutations of these residues, alone or combined, or the pharmacological inhibition of NUPR1 by ZZW-115, prevented the inhibitory action of NUPR1 on PARP1 activity.

We then treated MiaPaCa-2 cells with 5-FU to induce DNA damage, ZZW-115 and ZZW-115 in combination with 5-FU, and 5-FU in combination with Olaparib and measured the

PARylation effect by immunofluorescence using an anti-PAR antibody. Treatment with ZZW-115 or 5-FU induces 77.5 ± 7.57 and 52.1 ± 3.37 PARylation foci/nuclei respectively, whereas their combination shows a synergic effect (167 ± 12.1). These results were confirmed by western blot by using MiaPaCa-2 cells treated with 5-FU, ZZW-115 and olaparib, alone or in combination. As showed in Fig. 2d, treatment with 5-FU increased PARylation

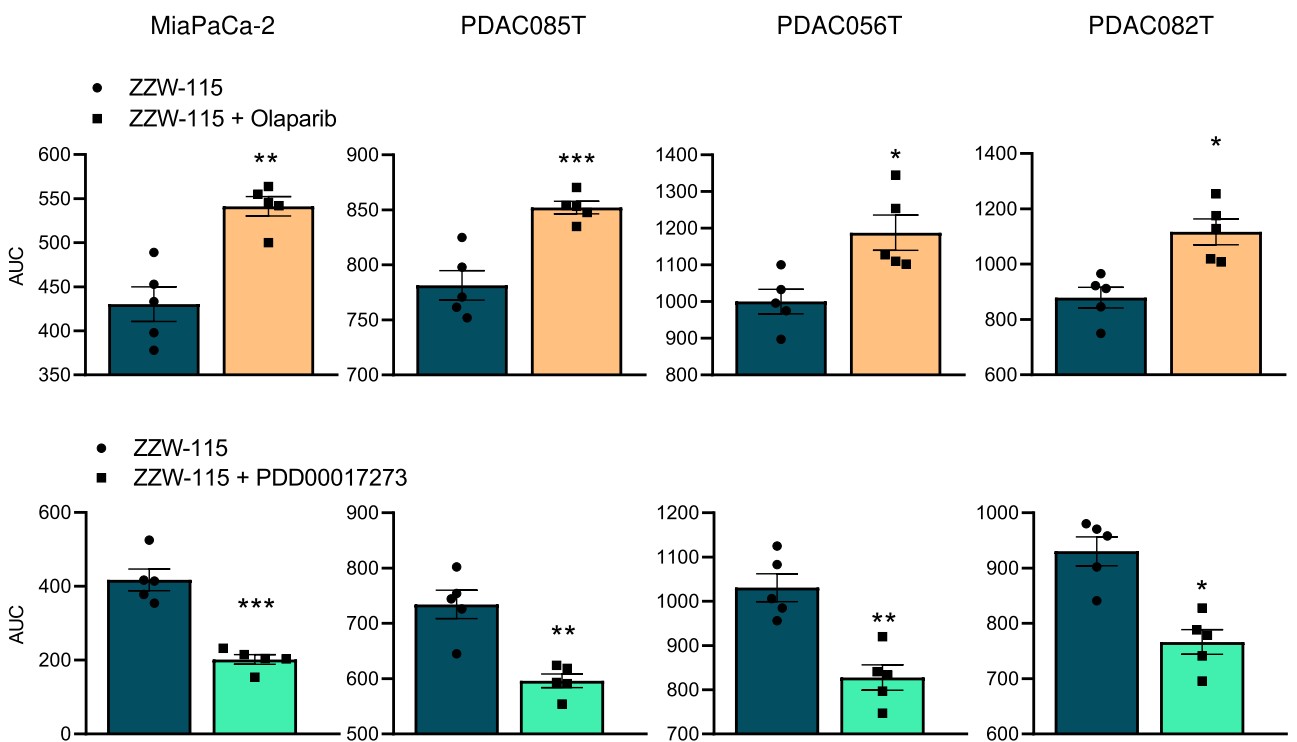

**Fig. 3 NUPR1 inhibition-induced cell death was rescued by PARP1 inhibitor and enhanced by PARG inhibitor.** Chemogram assays were done on 4 pancreatic cancer cell lines (MiaPaCa-2, PDAC085T, PDAC056T, and PDAC082T) with increasing concentrations of ZZW-115 in the presence or absence of olaparib (25 μM) or PDD00017273 (1 μM) for 24 h (n = 5). Curves were plot in GraphPad software and the area under the viability curve (AUC) was calculated by integration. Data represent mean ± SEM. Student's 2-tailed unpaired t test were used, *P < 0.05; **P < 0.01; ***P < 0.001; ****P < 0.001.

whereas, as expected, olaparib inhibited completely this effect. Interestingly, ZZW-115 increases the PARylation, which is also sensitive to olaparib, but extraordinarily, ZZW-115 treatment induced a synergic effect when it was combined with 5-FU. Altogether, our results strongly suggest that NUPR1 acts as an inhibitor of the PARP1 activity and its inhibition with ZZW-115 promotes hyperPARylation under stress conditions.

**Cellular NAD$^+$ concentration is decreased upon ZZW-115 treatment.** PARP1 consumes NAD$^+$ as substrate for PARylation. Deficit of NAD$^+$ could be responsible of the cell death occurring during hyperPARylation. We measured the NAD$^+$/NADH ratio in 5-FU treated cells in combination with olaparib or ZZW-115 to confirm that PARP1 activity is modulated. In agreement with PARylation effects, treatment with ZZW-115 or 5-FU decreases the NAD$^+$/NADH ratio. Combination of ZZW-115 with 5-FU induces a most important decrease of NAD$^+$/NADH ratio which is completely rescued by the co-treatment with olaparib (Fig. 2e).

**Cell death induced by ZZW-115 is rescued by olaparib and improved with PARG inhibitor.** NUPR1 inhibition by ZZW-115 induces cell death with an IC50 of around 1 μM in primary pancreas cancer cells, as previously described, and partially rescued by necrostatin-1 and Z-VAD-FMK[10]. Because NUPR1 inhibition is accompanied of hyperPARylation and this hyperPARylation could be responsible of cell death, we evaluated the effect of the inhibitor of PARP1, olaparib, and of the inhibitor of dePARylation, PDD00017273, on cell death induced by ZZW-115 on four primary pancreatic cancer cells. To do this, cells were treated with increasing concentrations of ZZW-115 in combination with constant concentration of olaparib (25 μM) or PDD00017273 (1 μM). Remarkably, the cell death effect of ZZW-115 was systematically inhibited by olaparib and, on the contrary,

PDD00017273 treatment improves the cell death effect (Fig. 3). We used flow cytometry after co-labeling cells with annexin V and PI to validate the previous observations (Supplementary Fig. 3). These results strongly suggest that cell death induced by NUPR1 inhibition by ZZW-115 involves the level of PARylation.

**Mitochondrial changes induced by ZZW-115 are rescued by olaparib.** NUPR1 inactivation induces a strong mitochondrial dysfunction[10,16,17]. Here, we studied the activity of the mitochondria after treatment with ZZW-115 alone or in combination with olaparib. As expected, ZZW-115-treatment induces a strong decrease of the oxygen consumption rate (OCR) which is rescued by olaparib in a dose-dependent manner (Fig. 4a). We then investigate the effect of olaparib treatment on mitochondria of MiaPaCa-2 cells treated with ZZW-115. Using the MitoTracker red to determine the cellular mitochondrial network, we observed that ZZW-115 treatment induces a strong disorganization, which agrees with our previous results[10,16,17]. Remarkably, this mitochondrial network disorganization is completely rescued by the treatment with olaparib (Fig. 4b) as well as other morphological changes induces by ZZW-115 such as the mitochondrial aspect ratio and the mitochondrial circularity (Supplementary Fig. 4).

Key features of PARP1-mediated cell death include mitochondrial membrane depolarization, ROS production and dependence on calcium signaling[18]. Therefore, we measured the mitochondrial membrane potential, mitochondrial superoxide levels, intracellular ROS levels and the cytosolic calcium concentration using MitoProbe™ TMRM, MitoSOX™ Red, CellROX™ Orange Reagent or Fluo-4-AM, respectively, in response to increasing amount of ZZW-115 in the presence or in the absence of olaparib. As presented in Fig. 4c, mitochondrial membrane potential decreases in a dose-response dependent manner, whereas the superoxide production (Fig. 4d), the intracellular level of ROS (Fig. 4e) and Ca$^{2+}$ cytosolic levels (Fig. 4f) increase in a dose-

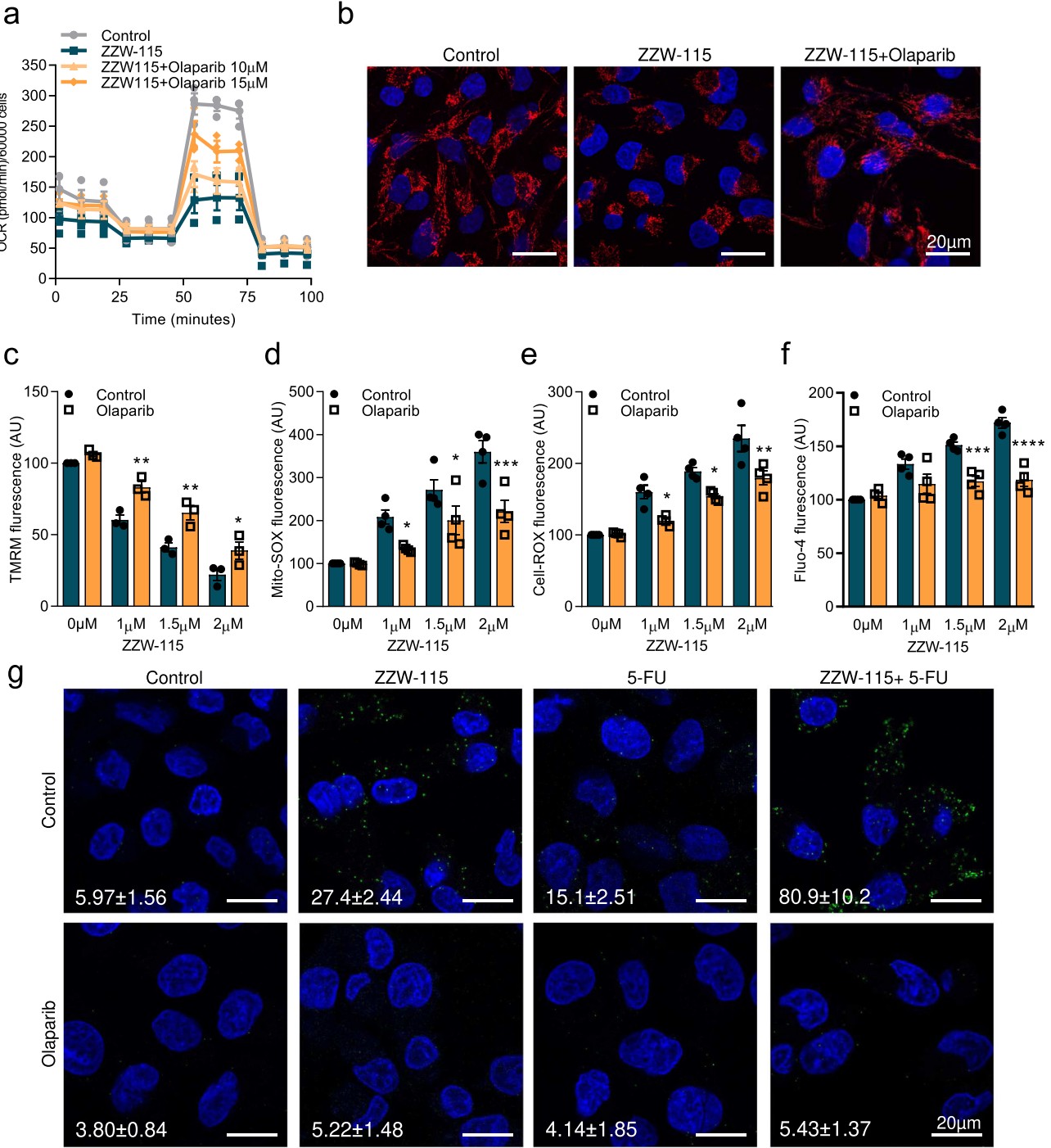

**Fig. 4 ZZW-115-induced mitochondrial dysfunction was reversed by PARP1 inhibitors. a** OXPHOS metabolism, reflected by oxygen consumption rate (OCR) levels were measured in MiaPaCa-2 cells after 24 h treatments. **b** MiaPaCa-2 cells were treated with ZZW-115 (1.5 μM) and olaparib (25 μM) for 24 h, then, loaded with MitoTracker Deep-Red FM and, after fixation, stained with DAPI. Flow cytometry analysis were carried out using MitoProbe™ TMRM (**c**), MitoSOX™ Red (**d**), CellROX™ Orange Reagent (**e**) or Fluo-4-AM (**f**) for analysis of the mitochondrial membrane potential, mitochondrial superoxide levels, intracellular ROS levels and the cytosolic calcium concentration, respectively, after 24 h of incubation with the drugs. Data represent mean ± SEM. Two-way, Sidak was used, *$P < 0.05$; **$P < 0.01$; ***$P < 0.001$; ****$P < 0.001$. **g** PLA was performed in MiaPaCa-2 cells in the presence or in the absence of ZZW-115 together or not with 5-FU in the presence or absence of olaparib. Mouse anti-PAR and rabbit anti-Mitofusin2 antibodies were used. A representative experiment is shown ($n = 3$). ImageJ was used to count the number of green dots. Mean ± SEM of foci/nucleus are included.

response manner. Importantly, these effects were rescued by the concomitant treatment with olaparib.

Besides that, PARP1 overactivation depletes cellular pool of $NAD^+$ and ATP, the product of this reaction, the PAR polymers, can also directly damage mitochondria. PAR polymers are normally accumulated in the nucleus, however, upon PARP1

overactivation, PAR polymers will translocate from the nucleus into the cytosol, inducing mitochondrial depolarization and opening of the mitochondrial permeability transition pore[19]. To address if, under ZZW-115 treatment, the PAR polymers translocate from the nucleus to the cytosol and interact with the mitochondria, we performed a PLA assay by using anti-PAR

and anti-Mitofusin2 antibodies. As it is shown in Fig. 4g, ZZW-115 treatment induces PLA positive signals from 5.97 ± 1.56 foci/cell in control cells to 27.4 ± 2.44 foci/cell in ZZW-115 treated cells. Remarkably, treatment with 5-FU shows 15.1 ± 2.51 foci/cell, but this signal increased to 80.9 ± 10.2 when combined with ZZW-115 indicating a strong synergic effect. In addition, specificity of this signal was demonstrated by the fact that these signals disappear almost completely when cells were co-treated with olaparib.

Altogether, these results demonstrate that ZZW-115 induces morphological and functional changes in mitochondria that can be rescued by olaparib treatment, indicating that these mitochondrial dysfunctions are, at least in part, downstream of PARylation activity and PAR polymers.

**HyperPARylation is sensitive to antioxidants**. Previously, we have reported that ZZW-115 induces ROS accumulation and mitochondrial disruption dependent of ROS accumulation, by disrupting the oxidative stress defense and inducing ROS-dependent cell death. As ROS are responsible of DNA damage and induce PARP1 activity, we studied the effect of the anti-oxidant inhibitor of ferroptosis, Ferrostatin-1, the antioxidant, N-acetylcysteine (NAC), and Mitoquinone on PARylation induced by ZZW-115 in MiaPaCa-2 cells to understand if there is a positive feedback between ROS accumulation and PARP1 activation upon NUPR1 inhibition. Cells were treated with 1.5 μM of ZZW-115 alone or in combination with Ferrostatin-1 (1 μM), NAC (20 mM) or Mitoquinone (0.5 μM) for 24 h and PARylation was measured by immunofluorescence with an anti-PAR monoclonal antibody. We observed a strong increase in PARylation in response to ZZW-115 which is almost completely rescued by each of both Ferrostatin-1, NAC and Mitoquinone treatments (Fig. 5a).

**The precursor of $NAD^+$ nicotinamide mononucleotide rescues mitochondrial activity and improves survival in ZZW-115 treated cells**. PARP1 uses oxidized NAD ($NAD^+$) as a substrate to ADP(ribosyl)ate its target proteins. The nicotinamide mononucleotide (NMN) is a precursor of $NAD^+$ which is transformed by the cells and utilized by PARP1. Here, we analyzed the activity of the mitochondria after treatment with ZZW-115 and NMN alone or in combination. As expected, ZZW-115-treatment induced a strong decrease of the oxygen consumption rate (OCR), which was rescued by 0.5 mM NMN (Fig. 5b). Then we performed a chemogram with increasing concentration of ZZW-115 in the presence or in the absence of NMN (0.5 mM). We found that the presence of NMN in the culture media improved the survival of the cells (Fig. 5c). These results suggest that, in addition to the effect of hyperPARylation, a decrease of $NAD^+$ is involved in the cell death induced by ZZW-115.

**Parthanatos is induced through a non-canonical pathway**. HyperPARylation induces a specific cell death named parthanatos[20]. The PAR polymer signaling to mitochondrial apoptosis-inducing factor AIF is the key event initiating the deadly crosstalk between the nucleus and the mitochondria in parthanatos. AIF release from the mitochondria and translocation to the nucleus is the commitment point for classical parthanatos. Thus, we studied if the cell death we observed in response to ZZW-115 could be mediated by AIF nuclear translocation. To analyze this hypothesis, we treated the cells with ZZW-115 alone or in combination with olaparib and studied the localization of AIF by immunofluorescence with a polyclonal antibody directed against AIF. Surprisingly, although the treatment with ZZW-115 induces a dramatic modification of the mitochondrial network,

AIF remains completely limited into the mitochondria with no release to the nucleus (Fig. 5d). Our results suggest that cell death induced by hyperPARylation is mediated by a non-canonical pathway.

**Cell death driven by genetic inhibition of NUPR1 is rescued by olaparib**. In order to demonstrate that the results presented above are the consequence of the inhibition of NUPR1 by ZZW-115, we performed a genetic downregulation of its expression by using a specific NUPR1-siRNA transfected to MiaPaCa-2 cells (Fig. 6a). First, we analyzed the cell viability after 5-FU and olaparib treatment. As expected, genetic inhibition of NUPR1 decreased cell viability, compared to control cells and 5-FU induced a decreasing on cell viability in both siRNA-transfected cells (control and NUPR1). Interestingly, olaparib, significantly increased cell viability in NUPR1-depleted cells (Fig. 6b). In addition, we measured the ATP levels in cells and found a correlation between the levels of ATP and the cell death effect. We also observed that PARP1 inhibition by olaparib can prevent in part the ATP depletion induced by NUPR1 inhibition (Fig. 6c). Moreover, we tested the PARP1 activity in cells by measuring the PARylation foci/nuclei. Genetic inhibition of NUPR1 increased the number of PARylation foci/nuclei from 5.02 ± 1.25 in siControl to 18.57 ± 3.48 in siNUPR1 treated cells. Interestingly, treatment of NUPR1-depleted cells by 5-FU induced a synergic effect in the accumulation of PARylation foci per cell from 38.79 ± 8.57 in siControl to 94.28 ± 7.59 in siNUPR1 treated cells. Finally, a strong decrease was found when cells were incubated in the presence of olaparib (Fig. 6d). These results confirm that: (i) genetic inhibition of NUPR1 induces PARP1 overactivation, with depletion in ATP levels and cell viability; and (ii) these effects could be rescued by the PARP1 inhibitor olaparib.

## Discussion
In this work we report that NUPR1 binds to PAPR1 and represses its overactivation in response to stress. We found that inhibition of NUPR1 with ZZW-115 induced a cellular hyperPARylation, which could be reversed by olaparib, but also by Ferrostatin-1 and NAC, followed of a strong mitochondrial failure. However, besides the involvement of PARP1 by the accumulation of PAR and the translocation of the polymers to the mitochondria, the parthanatos cell death induced by ZZW-115 does not involve the nuclear translocation of AIF and it is almost completely rescued by supplementing the media with the $NAD^+$ precursor NMN.

NUPR1 has been described as participating in many processes associated with cancer, including cell cycle regulation and apoptosis, senescence[21], autophagy[22], cell migration and invasion, development of metastases[9] and more recently in ferroptosis[23,24]. Importantly, NUPR1 received significant attention due to its role in development and progression of pancreatic ductal adenocarcinoma (PDAC)[25,26] as well as in hepatocellularcarcinoma (HCC)[27], non-small cell lung cancer[28], cholangiocarcinoma[29], glioblastoma[30], multiple myeloma[31], osteosarcoma[32], and more recently in ovarian cancer[32]. These results prompted us to develop a small compound inhibitor of NUPR1 to be used for treating cancers. Unfortunately, NUPR1 is an intrinsically disordered protein (IDP)[33] and as a consequence, a high throughput screening, based on the principles that are applied to well folded proteins for selection of inhibitors, cannot be applied for NUPR1. Consequently, we developed a multidisciplinary strategy and found that ZZW-115 binds to Ala33 and Thr68 residues of NUPR1, and shows dramatic anticancer effects in animal models of PDAC and HCC[10,17]. This anticancer effect is mediated, at least in part, by inducing cell death by apoptosis, necroptosis and ferroptosis since they are partially rescued by Z-VAD-FMK,

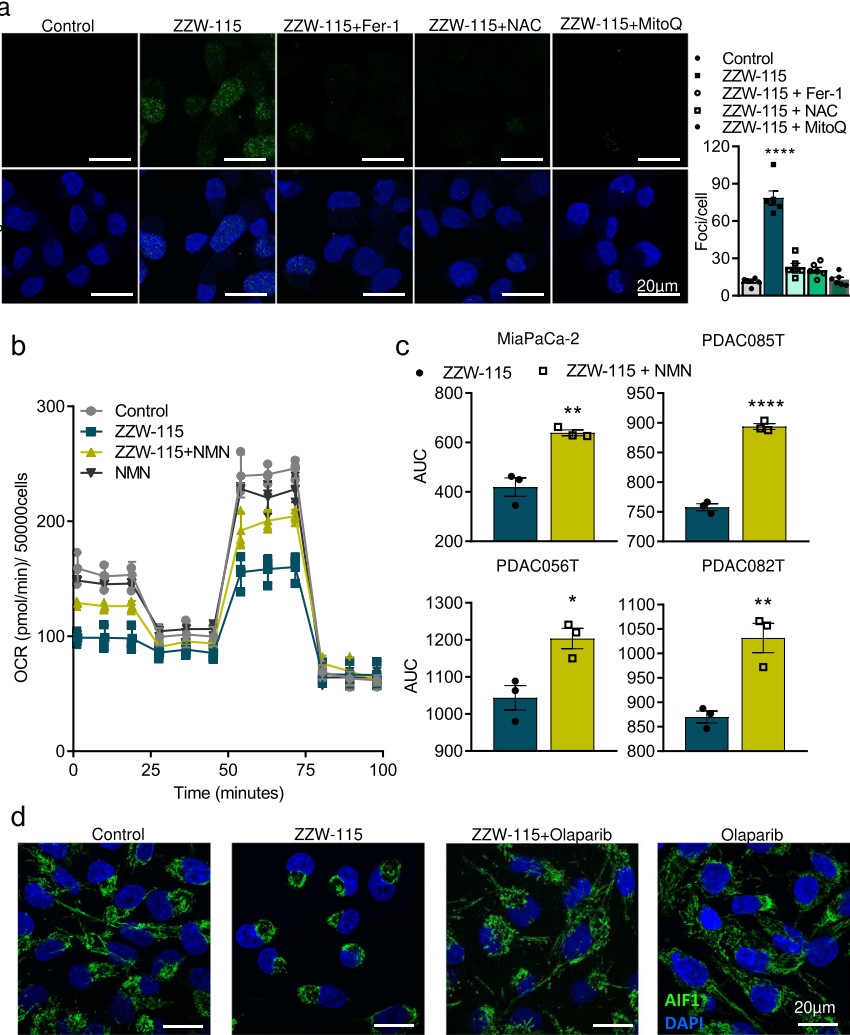

**Fig. 5 ZZW-115-induced cell death is rescued by antioxidants and NMN but not promoted by AIF translocation. a** MiaPaCa-2 cells were treated with ZZW-115 at 1.5 μM in isolation or in the presence of Ferrostatin-1 (1 μM), N-acetylcysteine (20 mM) or Mitoquinone (0.5 μM) for 24 h. Mouse anti-PAR and Alexa 488-labeled goat anti-mouse secondary antibody were used. A representative experiment is shown ($n = 3$). ImageJ was used to count the number of green dots ($n = 3$). Data represent mean ± SEM. One-way ANOVA, Dunnett correction; *$P < 0.05$; **$P < 0.01$; ***$P < 0.001$; ****$P < 0.001$. **b** OXPHOS metabolism, reflected by oxygen consumption rate (OCR) levels were measured in MiaPaCa-2 cells after 24 h treatment with ZZW-115 (1.5 μM) and NMN (0.5 mM) ($n = 3$). **c** Chemogram assays were done on 4 pancreatic cancer cell lines with increasing concentrations of ZZW-115 in the presence or absence of NMN (0.5 mM) for 24 h ($n = 3$). Curves were plot in GraphPad software and the area under the viability curve (AUC) was calculated by integration. Data represent mean ± SEM. Student's 2-tailed unpaired $t$ test was used, ***$P < 0.001$; ****$P < 0.001$. **d** Immunofluorescence was performed in MiaPaCa-2 cells in the presence or in the absence of ZZW-115 at 1.5 μM, olaparib 25 μM or the combination of ZZW-115 with olaparib. Rabbit anti-AIF and Alexa 488-labeled goat anti-rabbit secondary antibody were used. A representative experiment is shown ($n = 3$).

necrostatin-1, and ferrostatin-1, respectively. One of the most intense cellular effects that we observed when NUPR1 is genetically inactivated[16], or cells were treated with ZZW-115[10], is a mitochondrial catastrophe, accompanied with an intense decreases of OCR and ATP synthesis and a strong increase in ROS. We hypothesize that the multiple cell deaths triggered by NUPR1 inactivation could be explained by this mitochondrial dysfunction, large ROS overproduction and loss of ATP production. After these irreversible events, cells would activate different cell death pathways concomitantly. However, we did not find a rational explanation for the mitochondrial catastrophe induced as a consequence of the genetic inactivation of NUPR1 or the treatment with ZZW-115 until present.

We have also shown that PARP1 binds to NUPR1 at the same regions as ZZW-115 and the other proteins[9–11,13] by using two different pieces of evidence. First the NUPR1 mutants at positions Thr68 and Ala33 showed a large decrease in binding to PARP1 in

our in cellulo experiments. And second, the presence of ZZW-115 hampered the binding to NUPR1. Furthermore, we also showed in vitro that the binding seemed to alter the secondary structure of at least one of the proteins. Since PARP1 is a 1010-residue long protein and NUPR1 only contains 82 residues, we hypothesize that the protein changing its conformation is NUPR1. Usually NUPR1 remains disordered upon binding to any of its partners: either small molecules[9–11,15,16] or proteins[13,14], and we have only observed such large changes when NUPR1 is bound to plakophilin[12].

Interestingly, in this work we demonstrated that the hyper-PARylation induced by ZZW-115 treatment affects the mitochondrial network and function, since it is completely reversed by the treatment with olaparib, an efficient inhibitor of the PARP1 activity. In agreement, we also demonstrate that the cell survival is improved by olaparib and worsened by inhibiting dePARylation, indicating that the intracellular level of PARylation control

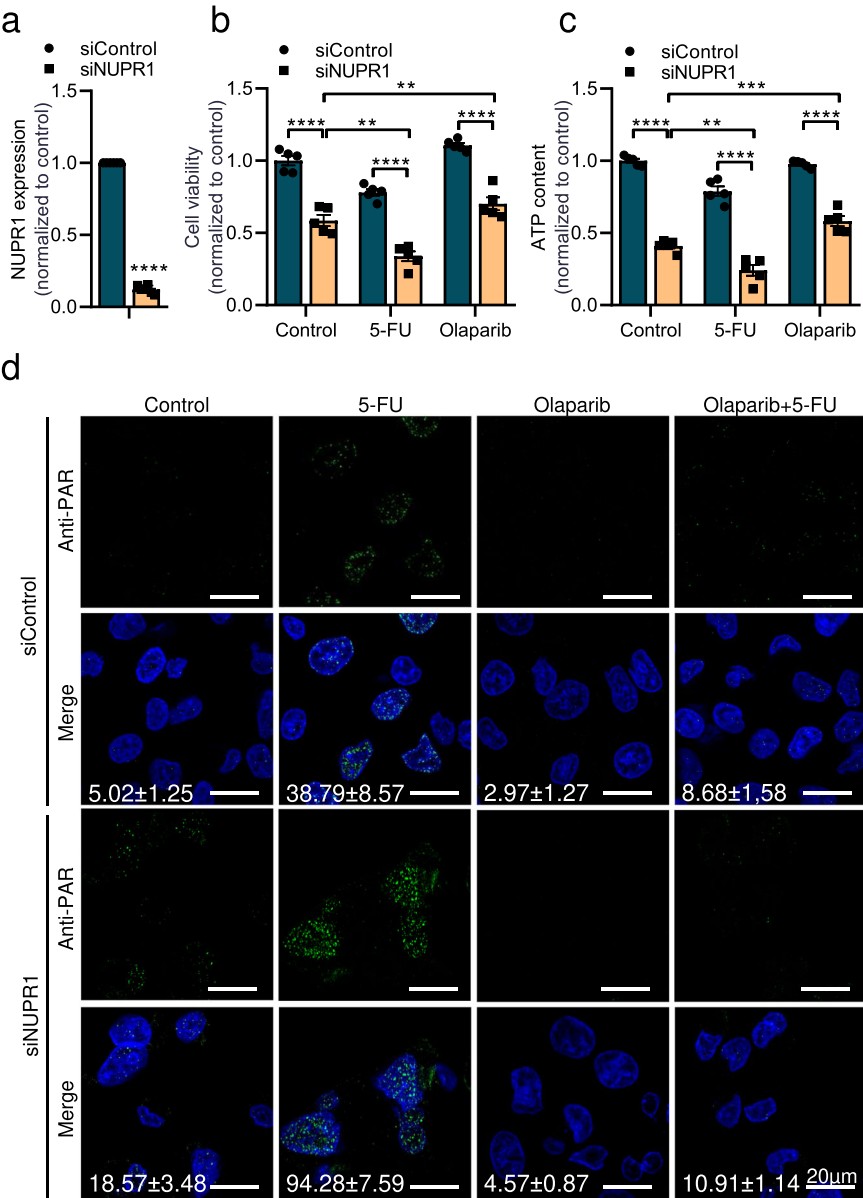

**Fig. 6 Olaparib prevented cell death and ATP level decrease in NUPR1-depleted cells.** MiaPaCa-2 cells were transfected with siControl or siNupr1 for 48 h. **a** Total RNA was extracted to monitor the mRNA level of NUPR1 by RT-qPCR 48 h post-transfection. Cells were treated with 5-FU or olaparib for 24 h. **b** Cell viability and (**c**) ATP levels were measured. Data represent mean ± SEM. Two-way, Sidak was used, *$P < 0.05$; **$P < 0.01$; ***$P < 0.001$; ****$P < 0.001$. **d** Immunofluorescence with a mouse anti-PAR and Alexa 488-labeled goat anti-mouse secondary antibodies were used. A representative experiment is shown ($n = 3$).

the cell survival during stresses. Remarkably, the cytotoxic effect of ZZW-115 does not involve nuclear translocation of AIF but it is reversed by the treatment with the $NAD^+$ precursor NMN, indicating that the cell death induced by this treatment is, contrary to expected, a non-canonical parthanatos. Importantly, some NUPR1-dependent effects are also involved in the resistance to some anticancer drugs[34] although its molecular explanation remains unknown. Based on our present observations we can hypothesize that during the cellular stress induced in response to the anticancer treatments, the NUPR1 activation could orchestrate a negative feedback with PARP1, limiting the hyperPARylation and its consequent mitochondrial catastrophe-dependent cell death.

Interestingly, olaparib was able to decrease ROS levels in ZZW-115-treated cells, meaning that cellular and mitochondrial ROS

accumulation can be prevented by the inhibition of PARP1 upon ZZW-115 treatment. However, we wanted to determine if there is a positive feedback loop of PARP1 overactivation by the accumulation of ROS. To our surprise, antioxidants agents such as Ferrostatin-1, NAC or Mitoquinone are capable of inhibiting the hyperPARylation, suggesting that ROS and mitochondrial ROS accumulation, upon ZZW-115, stimulates the PARP1 activity and therefore generates a positive feedback. The molecular model which we propose consists in an activation of the PARP1 activity in response a stress, which in the presence of the NUPR1 inhibitor ZZW-115 promotes the production of ROS, as a consequence of a mitochondrial dysfunction, and therefore a chronic stimulation of PARP1. Consequently, the hyperPARylation, resulting of the chronic activation of PARP1, induces the mitochondrial catastrophe and a cell death mediated by the withdrawal of the $NAD^+$.

Inhibition of NUPR1 by ZZW-115 could be a promising strategy to treat several tumors. However, based in the present results, inhibition of NUPR1 will not be applicable to tumors with homologous repair defects such as those developed on patients with germinal BRCA mutations in whose olaparib in combination with DNA-damaging agents is the indicated therapeutic strategy[35]. Thus ZZW-115 might be administered as a personalized therapy in patients without BRCA mutation.

## Methods

**Cell lines and cell culture**. MiaPaCa-2 cells were obtained from American Type Culture Collection (ATCC, USA) and cultured in Dulbecco's modified Eagle's medium (DMEM) containing 10% fetal bovine serum (Lonza, Basel, Switzerland) in an incubator with 5% $CO_2$ at 37 °C. PDAC primary cell cultures were obtained from xenografts as previously described[36]. Cells were cultured in serum-free ductal media (SFDM) at 37 °C in a 5% $CO_2$ incubator. Cell were tested for mycoplasma contamination.

**Flag- and GFP-NUPR1 coimmunoprecipitations and LC-MS/MS analysis**. The experimental set-up was the same as described previously[11]. Briefly, MiaPaCa-2 cells, expressing Flag-NUPR1 or GFP-NUPR1 or their controls, were plated in 10 $cm^2$ dishes. When MiaPaCa-2 cells expressing Flag-NUPR1 or GFP-NUPR1 reached 70% confluence, they were treated for 24 h and lysed. Equal amounts of total protein were used to incubate with 30 µL of anti-Flag M2-coated beads (MilliporeSigma, F3165) or GFP-Trap Agarose (Chromotek, GTA-10). Beads were then washed 3 times, and proteins were eluted using ammonium hydrogen carbonate buffer containing 0.1 µg/µL of Flag peptide (MilliporeSigma, F3290). Eluted proteins were collected and loaded on NuPAGE 4–12% Bis-Tris acrylamide gels according to the manufacturer's instructions (Invitrogen). Protein-containing bands were stained with Imperial Blue (Pierce), cut from the gel, and digested with high-sequencing-grade trypsin (Promega) before MS analysis. MS analysis was carried out by LC-MS/MS using an LTQ-Velos-Orbitrap or a Q Exactive Plus Hybrid Quadrupole-Orbitrap (Thermo Fisher Scientific) coupled online with a nanoLC Ultimate3000RSLC chromatography system (Dionex). Raw files generated from MS analysis were processed using Proteome Discoverer 1.4.1.14 (Thermo Fisher Scientific).

**NUPR1 coimmunoprecipitations**. NUPR1 and PARP1 recombinant proteins were mixed 1:1 (2 µg of each protein). Co-IP was performed using Dynabeads® and anti-NUPR1 antibody according to the manufacturer's protocol. Complexes were eluted by using an elution buffer and heating the samples for 10 min at 70 °C. Finally, the supernatant was loaded in SDS-PAGE and then evaluated by Western blotting.

**Western blotting**. Proteins were resolved by SDS-PAGE, and transferred to nitrocellulose membranes (GE Healthcare) for 1 h. Then, membranes were blocked 1 h at room temperature with TBS (Tris buffered saline solution) and 5% BSA, and blotted overnight in TBS 5% BSA containing primary antibodies Anti-Poly (ADP-Ribose) Polymer antibody (1:500, mouse, Abcam, ab14459), anti-GFP 1:500, (rabbit, Invitrogen, A-11122) anti-Flag (mouse, MilliporeSigma, F1804), anti-PARP1 (1:500, Cell Signaling, 9532), anti-NUPR1 (rabbit, homemade) or anti-β-actin (1:500, mouse, Sigma, A5316). After extensive washes in TBS 0.1% Tween-20, membranes were incubated 1 h at room temperature with HRP-conjugated secondary antibodies at 1:5000 before being revealed with Enhanced chemoluminescence (ECL). Acquisition was performed with a Fusion FX7 imager (Vilber-Lourmat). Uncropped blots are available in the Supplementary Fig. 6.

**Proximity Ligation Assay (PLA)**. Fifty thousand MiaPaCa-2 cells were seeded on coverslips. For some experiments, cells were transfected with 2 µg of plasmid DNA (PARP1-Flag and NUPR1-GFP) using Lipofectamine 3000 Transfection Reagent (Thermo Fisher Scientific). Twenty-four hours later, cells were treated. At the end of the experiment cells were washed in PBS, fixed, and permeabilized (0.02% Triton X-100 in PBS) before immunostaining with Duolink In Situ (MilliporeSigma) following the manufacturer's protocol. Anti-NUPR1 (rabbit, homemade) and anti-Flag (mouse, MilliporeSigma, F1804), Mouse anti-PAR (mouse, ab14459) and rabbit anti-Mitofusin2 (Cell Signaling, 9482) or mouse anti-Flag (mouse, MilliporeSigma, F1804) and rabbit anti-PARP1 (Cell Signaling, 9532) antibodies were used. Image acquisition was carried out confocal microscope, LSM 880 (×63 lens) controlled by Zeiss Zen Black. ImageJ 1.53C (NIH) was used to count the number of green foci.

**Protein expression and purification**. Wild-type NUPR1 and Thr68Gln, Ala33Gln, Thr68Gln/ Ala33Gln NUPR1 mutants were produced and purified in LB media as described before[13]. The strain used for protein expression was NEB® Stable Competent E. coli (High Efficiency) C3040H. A denaturing SDS-PAGE/ coomassie blue is provided in the supplementary information (Supplementary Fig. 5). Pure PARP1 at concentrations of 1.15 mg/ml were obtained from Sino Biological (China).

**PARP1 activity assay**. PARP1 activity was measured with the PARP1 Enzyme Activity Assay Kit (EMD Millipore Corporation, 17-10149) according to the manufacturer's instructions. PARP1 enzyme, β-NAD, activated DNA, and recombinant nicotinamidase enzyme were combined and incubated for 30 min in the presence of WT-NUPR1, recombinant mutants of NUPR1 (Thr68Gln, Ala33Gln, Thr68Gln/Ala33Gln) or ZZW-115. During the incubation, the activated DNA triggers PARP1 to produce poly(ADP-ribose) and nicotinamide. In a secondary reaction, the nicotinamidase enzyme converts the nicotinamide into nicotinic acid and $NH_3^+$ (free ammonia). To generate a signal for readout, the developer reagent is added and the signal was read using a TECAN infinitive plate reader.

**Cell viability**. Cells were plated in 96-well plates (5000 cells/well). Twenty-four hours later, the media were supplemented with various concentrations of ZZW-115 (synthetized as previously described[10]) in the presence or absence of olaparib (Sellechem), PDD00017273 (Merk) or NMN (Merk), and were incubated for an additional 24 h period. Cell viability was estimated by crystal violet. Medium was discarded; cells were fixed with 1% glutaraldehyde solution, washed with PBS and stained with 0.1% crystal violet solution in 70% methanol. After discarding the crystal violet solution, cells were washed with PBS three times and 1% SDS solution was added to solubilize the stain. Absorbance was read at 590 nm on Epoch™ Microplate Spectrophotometer (Biotek). AUC (area under the curve) values were calculated by nonlinear regression curves with a robust fit using GraphPad software.

**Determination of NAD+/NADH ratio**. Cells were plated in 96-well plates (15,000 cells/well). Twenty-four hours later, the media were supplemented with ZZW-115, olaparib and or 5-FU. $NAD^+$ and NADH levels were determined by using Promega $NAD^+$/NADH-Glo kit (G9071) using manufacturer's instructions. After treatment, media was exchanged to PBS and cells were lysed in 1 % DTAB (Dodecyltrimethylammonium bromide) and 0.2 M NaOH. Cell lysate was immediately divided into two 50 µl aliquots. One aliquot, to measure NADH was left unmodified, and the other aliquot, to measure $NAD^+$, was supplemented with 25 µl of 0.4 M HCl. Both aliquots were heated at 60 °C for 20 min to destroy reduced or oxidized nucleotides and then, incubated 10 min at room temperature. The first one was complemented with 50 µl of HCl/Trizma solution and the second one with 25 µl of Trizma base. Finally, NAD/NADH-Glo™ Detection Reagent was added to the samples and they were incubated at room temperature for 45 min. $NAD^+$ and NADH levels were determined using a Tristar multimode microplate reader (Berthold).

**Immunofluorescence of cultured cells**. Cells were seeded in 24-well plates on coverslips and treated with ZZW-115, olaparib and of 5-FU. After fixation in PBS 4% FPA, cells were incubated with the following primary antibodies at 1:100 dilution: mouse anti-Poly (ADP-Ribose) Polymer (Abcam, ab14459), rabbit anti-AIF (1:100, Cell Signaling, 4642). After washing steps, samples were incubated in the presence of secondary antibodies at 1:200 dilution (goat anti-mouse Alexa Fluor 488, A28175, or goat anti-rabbit Alexa Fluor 488, A27034, both from Thermo Fisher Scientific). DAPI (D1306, Thermo Fisher Scientific) was used to stain the nucleus. Image acquisition of Alexa Fluor 488-derived fluorescence and DAPI staining was performed using an LSM 880 controlled by Zeiss Zen Black 63× lens. Analysis and measurement of both channels was conducted by using the ImageJ 1.53C software.

**Measurement of ROS and mitochondrial ROS**. Cells were seeded at $8 \times 10^4$ cells per well in 24-well plates. Next day, cells were treated with olaparib and indicated concentrations of ZZW-115 for 24 h. After that, cells were incubated with 5 µM CellROX Green Reagent (C10444, Thermo, USA) or 10 µM MitoSOX Red (M36008, Thermo, USA) at 37 °C for 30 min in the dark. Then, the unincorporated dye was removed by washings with prewarmed PBS. Samples were then harvested by accutase (Thermo Fisher), centrifuged at 1500 rpm for 5 min and the pellets were resuspended in 200 µL prewarmed HBSS (Hanks' Balanced Salt Solution, Gibco, Life Technologies) for flow cytometry. Ten thousand events per sample were collected in a MACSQuant-VYB (Miltenyi Biotec, Surrey, UK), and data were analyzed with FlowJo 10.7.1 software.

**Mitochondrial membrane potential assay**. Mitochondrial membrane potential assay was performed using MitoProbe TMRM Assay Kit (M20036, Invitrogen) following the manufacturer's protocol. After 24 h of incubation in presence of ZZW-115 and or olaparib, cells were dissociated using accutase (Thermo Fisher) and resuspended in 200 µL PBS at the density of $1 \times 10^6$ cells/mL. Then, we added 1 µL of 20 µM stock TMRM reagent solution to the cells and incubate for 30 min at 37 °C, 5% $CO_2$. Data were analyzed on flow cytometry with 561 nm excitation. Ten thousand events per sample were collected in a MACSQuant-VYB (Miltenyi Biotec, Surrey, UK). Data analysis was performed using the FlowJo 10.7.1 software.

**Cytosolic calcium concentration**. Cells were seeded at $8 \times 10^4$ cells per well in 24-well plates. After 24 h of treatment, cells were preincubated with Fluo-4 AM (Molecular Probes) for 30 min at 37 °C. Then, cells were washed and resuspended in HBSS for flow cytometry analysis. Ten thousand events per sample were collected in a MACSQuant-VYB (Miltenyi Biotec, Surrey, UK) and data analysis from flow cytometry was performed using FlowJo 10.7.1 software.

**Annexin V/PI staining**. Cells were collected after incubation for 24 h of treatment. Cells were washed and then detached with Accutase, and resuspended in annexin-binding buffer. Pacific-Blue annexin V (5 µl, BioLegend) was added to the cell suspension and incubated for 15 min. Before analysis by flow cytometry, propidium iodide (5 µl, Miltenyi Biotec) was added to the suspension. A MACSQuant-VYB (Miltenyi Biotec) was used to collect 10,000 events per sample. Data analysis was performed using FlowJo 10.7.1 software.

**Mitochondrial network**. Localization of the mitochondrial network was assayed by incubation of cells in the presence of MitoTracker Deep-Red FM (200 nM, Molecular Probes) at 37 °C for 30 min. Then, cells were washed twice with PBS and fixed with 4% paraformaldehyde. Finally, samples were mounted using the Prolong Gold antifade reagent with DAPI (Thermon Fisher). Confocal images were acquired using a confocal microscope LSM 880 controlled by Zeiss Zen Black 63x lens.

**Measurement of mitochondrial oxidative phosphorylation (OXPHOS)**. Thirty-five thousand cells/well were plated at 24-well plate (Seahorse) and incubated overnight in Standard DMEM. Cells were treated with ZZW-115, olaparib or NMN at indicated concentrations for 24 h. The Oxygen Consumption Rate (OCR) (pmol O$_2$/min) was measured using the Seahorse Bioscience XF24 Extracellular Flux Analyzer (Agilent). Before the measurement of OCR, cells were incubated in XF assay medium OCR measurement is under basal conditions in response to 1 µM oligomycin, 0.25 µM carbonylcyanide p-(trifluoro-methoxy)phenylhydrazone (FCCP), 0.5 µM rotenone (MilliporeSigma), and 2-deoxyglucose (100 mM) addition (MilliporeSigma). Levels of OCR were normalized to the cell number by crystal violet.

**siRNA transfection**. Cells were plated at 70% confluence and INTERFERin reagent (Polyplus-transfection) was used to perform siRNA transfections, according to the manufacturer's protocol. Scrambled siRNA that targets no known gene sequence was used as a negative control. All assays were carried out 72 h post-transfection. The sequence of Nupr1-specific siRNA was siNUPR1-1 r(GGAG-GACCCAGGACAGGAU)dTdT.

**ATP assay**. 24 h post-transfection, MiaPaCa-2 cells were seeded at a density of 10,000 cells/well in 96-well plates. Cells were allowed to attach for 24 h and then treated for another 24 h. ATP production was monitored 72 h post-transfection using CellTiter-Glo (Promega G7571) assay. Data was normalized to the cell number.

**RT-qPCR**. RNA from cells was isolated using Trizol reagent (Invitrogen) and reverse-transcribed using Go Script (Promega) according to the manufacturer's instruction. Real-time quantitative PCR (RT-qPCR) was performed in a Stratagene MXPro-MX3005P using Promega reagents.

**Far-UV CD experiments**. Far-UV CD spectra were collected on a Jasco J810 spectropolarimeter (Jasco, Tokyo, Japan) with a thermostated cell holder, and interfaced with a Peltier unit at 25 °C. The instrument was periodically calibrated with (+)-10-camphorsulphonic acid. A path length cell of 0.1 cm was used (Hellma, Kruibeke, Belgium). All spectra were corrected by subtracting the corresponding baseline. The experiments were carried out as described[12]. Samples were the same used in fluorescence experiments.

**Fluorescence experiments**. Fluorescence spectra were collected on a Cary Varian spectrofluorometer (Agilent, Santa Clara, CA, USA), interfaced with a Peltier unit. All experiments were carried out at 25 °C. The experiments were carried out as described[12]. A 1-cm-path length quartz cell (Hellma, Kruibeke, Belgium) was used. Concentration of PARP1 was 2 µM and that of NUPR1 was 20 µM; samples at those concentrations were prepared containing isolated PARP1, isolated NUPR1 and a mixture of both. Experiments were performed at pH 7.5, in 50 mM Tris buffer.

**Statistics and reproducibility**. Statistical analyses were conducted by using the unpaired two-tailed Student $t$ test, or one-way ANOVA with Dunnett correction or two-way ANOVA with Sidak correction and d'Agostino pearson test. The results were expressed as the mean ± SEM of at least three independent experiments. A $p$ value of <0.05 was regarded as statistically significant. GraphPad Prism8 software was used. Mitochondrial circularity and mitochondrial aspect ratio were calculated on ImageJ 1.53C (NIH) with Mitochondrial Analyzer plugin.

**Reporting summary**. Further information on research design is available in the Nature Research Reporting Summary linked to this article.

## Data availability

The mass spectrometry proteomics data have been deposited to the ProteomeXchange Consortium via the PRIDE[37] partner repository with the dataset identifier PXD035236. The datasets generated during the current study and the processed mass spectrometry-based proteomics data are available in the Figshare repository, respectively: https://doi.org/10.6084/m9.figshare.20079917 and https://doi.org/10.6084/m9.figshare.20079926. Uncropped blots are available in the Supplementary Fig. 6.

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

## Author contributions

P.S-C., P.S., and J.L.I. proposed the concept and designed the experiments; P.S-C., C.H., X.L., Y.X., S.A., J.L.N., and L.C. carried out experiments; P.S-C., L.P., G.L., R.U., P.S., J.L.N., and J.L.I. analyzed and interpreted the results; P.S-C., G.L., R.U., P.S., J.L.N., and J.L.I. wrote the paper.

## Competing interests

J.I. is co-founder of PanCa Therapeutics and PredictingMed. P.S.C., Y.X., L.P., J.L.N., and J.I. are inventors of the Patent "NUPR1 INHIBITION FOR TREATING CANCER", Application number WO-2019229236-A1. The other authors declare no competing interests. This work was supported by La Ligue Contre le Cancer, INCa, Canceropole PACA, and INSERM. C.H. and X.L. are recipient of the predoctoral fellowship from China Scholarship Council (CSC).
