## [Peer Review File · Communications Biology]

Reviewers' comments:

Reviewer #1 (Remarks to the Author):

In this manuscript, Iovanna et al described and characterized NUPR1 as a new PARP1-binding protein and a PARP1-inhibiting protein. Authors previously identified a potent NUPR1 inhibitor ZZW-115 as a new anticancer therapeutic (with patent) and, based on the mitochondrial catastrophe, proposed a necroptosis as a mechanism of cell-death induced by ZZW-115. In this study, they moved forward with PARP1-linked function of NUPR1 and proposed a non-canonical parthanatos as a mechanism of ZZW-115-induced cell death. Overall, this study provides a new insight into the NUPR1 function and the PARP1-dependent cell death pathway, identified a new regulator of PARP1, and importantly provides a mechanistic basis for ZZW-115 as a cancer therapeutic. This reviewer believes this will be of general interest to audience in cancer biology and PARP biology area. However, there are some missing data that support the physical and functional interactions between NUPR1, ZZW-115, and PARP1. This reviewer recommends publication after revising these points.

Specific points;

Major points:

1. It is not clear whether recombinant NUPR1 (WT and mutants, in particular A33Q/T68Q) physically interacts with recombinant PARP1 in vitro. Authors should address this to claim NUPR1 as a PARP1-binding protein.
2. It is also important to show whether NUPR1 interacts with PARP1 or PAR (PARylated PARP1).
3. In Figure 2, it is recommended to show a dose-dependent inhibition of PARP1 at a fixed concentration of PARP1 by titrating NUPR1 (WT and mutants) +/- ZZW-115 to higher concentrations to show the degree of inhibition.
4. To conclude whether ZZW-115 involves non-canonical parthanatos, cellular data with ZZW-115 +/- caspase inhibitors should be provided.

Minor points:

1. Authors should describe the preparation of recombinant NUPR1 (WT and mutants) in methods, and show purity in supplementary information.
2. It would be helpful if authors show the chemical structure of ZZW-115 and its possible interaction with NUPR1 in supplementary figure.
3. The full name of 5-FU needs to be described in the manuscript for general audience/reader.
4. There seem to be many typos. Please correct.
Line 151: subtract? Substrate?
Line 205: antioxidant and inhibitor of ferroptosis, Ferrostatin-1,
Line 219: in the presence or not (absence?)
Line 349: Fifth?

Reviewer #2 (Remarks to the Author):

PAR (Poly-ADPribose) polymerase 1 (PARP1) activity is tightly regulated and multiples pathways are involved in its regulations in cancer cells. Parthanatos (a regulated cell death, RCD) is mediated by high level of PARP1 activity. In this manuscript, Santofimia-Castaño et al., demonstrated that nuclear protein 1 (NUPR1), a nuclear intrinsically disordered protein (IDP) of 82 amino acids, binds to PAR (Poly-ADPribose) polymerase 1 (PARP1) and inhibits PARP1 activity in pancreatic cancer cells. More importantly, the authors also found that the NUPR1 inhibitor ZZW-115 abolished the protective effect of NUPR1 as a PARP1 activity regulator, inducing strong hyperPARylation and cell death mediated by a strong mitochondrial dysfunction. This type of cell

death is a non-canonical parthanatos, which extends our understanding of NUPR1-regulated biological process in pancreatic cancer as well as the other cancers.

The experiments are well-designed and results are interesting. This reviewer has minor suggestions as follows:

1, The authors claimed that ZZW-115 and PARP1 are competing for the same binding site in NUPR1, involving Ala33 and Thr68. Thus, the mutations of Thr68Gln/Ala33Gln in NUPR1 prevent its inhibition on PARP1 activity as shown in Figure 2B. Does this mean these mutations Thr68Gln/Ala33Gln in NUPR1 loss its binding activity to PARP1? The authors may test their association by proximity ligation assay (PLA) or co-IP or pulldown in vitro.

2, NUPR1 inhibitor ZZW-115 also induces ferroptosis in a mitochondria-dependent manner, as previously reported in *Cell Death Discov* 2021;7:269. The authors would mention why so many different cell death pathways are connected to NUPR1 inactivation, including senescence in lung cancer and breast cancer (*Cell Death Dis* 2021 Feb 4;12(2):149. *Autophagy*. 2018;14(4):654-70.). Therefore, the authors may put their thoughtful insights of NUPR1 inactivation in Discussion section.

3, As an IDP and a transcriptional cofactor, NUPR1 may also show its function in gene regulation. It is clear that most of the PLA foci were nuclear location in Figure 1C, the PLA data (mouse anti-Flag and rabbit anti-NUPR1 antibodies). In this case, NUPR1 location may be performed in the presence of ZZW-115 in MiaPaCa-2 cells using rabbit anti-NUPR1 antibody alone, compared to the NUPR1-negative cancer cells.

4. NUPR1 activation could orchestrate a negative feedback with PARP1, limiting the hyperPARylation and its consequent mitochondrial catastrophe dependent cell death, which was reversed by olaparib (PARP1 inhibitor) or the NAD⁺ precursor nicotinamide mononucleotide (NMN). That is ROS-related signaling process, connecting the mitochondrial dysfunction to various cell death pathways. It has been known that NUPR1 is a stress-inducible protein and/or chemotherapy resistance-related protein, the authors might discuss this feature of its dependency such as on germinal BRCA mutations or more widely personal cancer treatment, in the Discussion section.

Reviewer #3 (Remarks to the Author):

Review summary on the "NUPR1 protects against hyperPARylation-dependent cell death" manuscript

Patricia Santofimia-Castaño and her coworkers investigate the protective role of NUPR1 against hyperPARylation-dependent cell death. Overall, the manuscript is well-written, the majority of the results are convincing. One of the strongest points of the manuscript is that the authors use their formerly developed small-compound (ZZW-115) that inhibits the activity of NUPR1. The authors use 4 cell lines to highlight their results, and the canonical points of the manuscript were confirmed by NUPR1 knockdown. The introduction and the results are clear and concise, and the discussion is articulate. However, I have some concerns about the experiments.

Major comments:

Materials and methods:

In the cell viability section, the authors state that "Medium was discarded". Due to medium removal, you discard detached dead cells as well, so you will only measure the remaining, theoretically viable cells. However, the crystal violet assay assumes that all cells attached to the plate are viable which is not always true. In this case, Figure 3 does not exclusively represent cell death, instead it may refer to cell cycle arrest as well. PARP1 inhibitors are strong cell cycle arrest inducers. I would suggest to confirm this result with another cell death detection method. For instance, using Propidium iodide - Annexin V double labelling followed by flow cytometry would have been a better choice. Or at least try to exclude the possibility of cell cycle arrest involvement

by BRDU or Propidium Iodide incorporation.

Measurement of OXPHOS. It won't turn out if the authors used normalization after measuring OXPHOS. Normalization in connection with Seahorses studies is a must especially when a treatment induces cell death. The detected lower oxygen consumption in response to ZZW-115 treatment might originate from lower cell viability and OCR values may merely represent cell death as seen by crystal violet assay. I would suggest to do normalization by measuring protein content after performing Seahorse. For example, dissolving the cells in NaOH or lysis buffer and performing a BCA assay. Another option is to present OCR/ECAR values on the graph regardless of normalization. However, it must be noted that OCR/ECAR might somewhat have a different meaning and the results should be interpreted accordingly.

Minor comments.

Introduction:

Line 86: "and consequently inhibits the nuclear translocation and therefore its nuclear activity". In my opinion the hampered nuclear translocation of a protein does not necessarily result in decreased protein activity. Although, this is true for NUPR1, I would omit „therefore“.

Results:

In Figure 2 and 5 I would add separately which statistical tests were used for Figure 5A and 5C and for Figure 2 A/B and Figure 2E. I assume it is not the same as stated in the Figure legend.

Figure 2D: The name of the housekeeping protein used for normalization is missing below anti-PAR.

Figure 3 might be misleading. If someone is not familiar with crystal violet staining, they won't find out if the graphs represent cell death or cell viability. I would at least mention in the figure legend what does the Y axis represent. I also don't see the increasing concentrations of ZZW-115 written in the figure legend. The same applies for Figure 5.C. It would have also been better in Figure 3 to use the "control" for the non-treated cells, the second group would have been the NUPR1 inhibited cells, the 3. one NUPR1 + olaparib and the 4. one NUPR1 + PDD00017273. Thereby, one could easily compare the extent of cell death among the different groups.

Line 175: "strong disorganization" referring to Figure 4B: Although, the morphological alteration of the mitochondrial network is obvious, rounded-like structures appear in response to ZZW-115 treatment, it would be nice to characterize the morphology of mitochondria. I would suggest to calculate for example mitochondrial circularity, form factor, aspect ratio etc. and add these parameters to the given figure. Images can be analyzed by the ImageJ software Mito-Morphology Macro among others. I would also check some mitochondrial dynamics regulator factors (Mfn1, Mfn2, Mff, Drp1, etc.) either by qPCR or by Western blot. Thereby, you could get an insight into the fission-fusion machinery and confirm the mitochondrial morphology with expressional changes.

I suggest to add the NUPR1 siRNA western blot picture either on Figure 6 or in the supplementary file and calculate the knockdown efficiency.

Figure 6B: It is a little confusing how you measured ATP content. Did you calculate ATP level from basal OCR by subtracting OCR values after oligomycin injection or did you perform ATP measurement by an ATP kit? It should be added in the M&M section.

The graphical abstract is well-made and easy to understand. Do you have any theory how mitochondrial stress leads to ROS production? Does mitochondrial ROS itself induce PARP1 activation? I would be nice to check in Figure 5A how PARP1 activation is affected upon treatment with a specific mitochondrial ROS scavenger (MitoTEMPO) to confirm if ROS with mitochondrial origin triggers PARP1 activation.

Materials and methods:

Line 340: Please define the manufacturer of the nitrocellulose membrane.
Line 352: Please define the permeabilization solution.
Line 356, 374 and 397: Please define the version number of ImageJ software.
Line 369 and 378: Please define the manufacturer of ZZW-115, olaparib, 5-FU, PDD00017273 and NMN.
Line 374: Please define the manufacturer of Epoch™ Microplate Spectrophotometer.
Line 387: Please define the manufacturer of Tristar multimode microplate reader.
Line 390: Please define the fixation solution.
Line 403: Please define the manufacturer of accutase solution.
Line 414 and 420: Please define the version number of FlowJo software.
Line 425: Please define the manufacturer of Prolong Gold antifade reagent with DAPI solution.
Line 430: Please define the manufacturer of Seahorse Bioscience XF24 Extracellular Flux Analyzer.

In the statistical analysis part, I would add which test was used to determine the normality of the groups in the case of ANOVA (Kolmogorov-Smirnov test, Shapiro-Wilk test etc.), as it affects if parametric or non-parametric test must be used. Please define the program and its version number used for statistical evaluation.

Seeded cell density are missing in some paragraphs in the methods section.

The meaning of the abbreviations sometimes is not obvious and they should be mentioned for example in the case of DTAB, 5-FU, HBSS, OXPHOS etc.

Evaluation:

Patricia Santofimia-Castaño and her coworkers' manuscript provides valuable data on the NUPR1 inactivation- and PARP1-mediated cell death involving mitochondrial dysfunction. The results embrace diverse methodological approach and they lay the foundation for further studies. However, I would like to ask the authors to address my concerns and comments.

Reviewers' comments:

Reviewer #1 (Remarks to the Author):

In this manuscript, Iovanna et al described and characterized NUPR1 as a new PARP1-binding protein and a PARP1-inhibiting protein. Authors previously identified a potent NUPR1 inhibitor ZZW-115 as a new anticancer therapeutic (with patent) and, based on the mitochondrial catastrophe, proposed a necroptosis as a mechanism of cell-death induced by ZZW-115. In this study, they moved forward with PARP1-linked function of NUPR1 and proposed a non-canonical parthanatos as a mechanism of ZZW-115-induced cell death. Overall, this study provides a new insight into the NUPR1 function and the PARP1-dependent cell death pathway, identified a new regulator of PARP1, and importantly provides a mechanistic basis for ZZW-115 as a cancer therapeutic. This reviewer believes this will be of general interest to audience in cancer biology and PARP biology area. However, there are some missing data that support the physical and functional interactions between NUPR1, ZZW-115, and PARP1. This reviewer recommends publication after revising these points.

Specific points;

Major points:

1. It is not clear whether recombinant NUPR1 (WT and mutants, in particular A33Q/T68Q) physically interacts with recombinant PARP1 *in vitro*. Authors should address this to claim NUPR1 as a PARP1-binding protein.

We agree with this comment and therefore we have performed new experiments to address this question. To confirm that NUPR1 is a PARP1 binding protein we have validated our previous data by demonstrating that the interaction took place *in vitro* by using two spectroscopic techniques: far-UV circular dichroism (CD) and fluorescence. Both confirmed that NUPR1 binds to PARP1. Results are included in the Figure 1D of the revised version of the manuscript. In addition, we have performed PLA assays with Flag-NUPR1 (WT, single mutants A33Q or T68Q, or double mutant A33Q/T68Q) and PARP1. These experiments have shown that single or double mutations decrease significantly the binding between NUPR1 and PARP1 as presented in Supplementary Figure 1B.

2. It is also important to show whether NUPR1 interacts with PARP1 or PAR (PARylated PARP1).

We find this comment very interesting; therefore we have completed our previous PLA data. In the Figure 1C we have challenged cells with ZZW-115, 5-FU or Olaparib. Our results had shown that: ZZW-115 was able to decrease the interaction; 5-FU, a DNA damage agent which activates PARP1, was able to increase the interaction, meaning that under stress, as previously was shown with the glucose starvation, NUPR1 and PARP1 interact better. In addition, 5-FU increases PARylated PARP1. Finally, Olaparib did not induce any change. All in all, we can conclude that NUPR1 can also bind PARylated PARP1. However, to validate this data, we have performed in the same conditions a PLA between NUPR1 and PAR (Supplementary Figure 1A). Only upon 5-FU treatment, which induces PARylated PARP1, positive interaction was found.

3. In Figure 2, it is recommended to show a dose-dependent inhibition of PARP1 at a fixed concentration of PARP1 by titrating NUPR1 (WT and mutants) +/- ZZW-115 to higher concentrations to show the degree of inhibition.

We thank this reviewer for her/his comment. NUPR1 WT results are shown in Figure 2A, mutants and ZZW-115 are shown in Supplementary Figure 2. NUPR1 WT is the only recombinant protein able to reduce PARP1 enzymatic activity. ZZW-115 showed a dose-dependent reduction on the inhibitory

effect of NUPR1 in PARP1 activity.

4. To conclude whether ZZW-115 involves non-canonical parthanatos, cellular data with ZZW-115 +/- caspase inhibitors should be provided.

We fully agree with the reviewer that studying alternative mechanism of cell death is necessary. In previous works we already used ZZW-115 in combination with Z-VAD-FMK, as well as other cell death inhibitors (DOI: 10.1172/JCI127223, DOI: 10.1038/s41420-021-00662-2 or DOI: 0.1016/j.canlet.2020.04.024). We found that Z-VAD-FMK was able to inhibit partially cell death upon ZZW-115-treatment. However, and very important, inhibition of caspase did not prevent mitochondrial catastrophe. In this regard, we found remarkable results indicating that the mechanism of cell death presented in this work, as well as key features such as the mitochondrial and metabolic catastrophe, are inhibited by the PARP1 inhibitor Olaparib.

Minor points:

1. Authors should describe the preparation of recombinant NUPR1 (WT and mutants) in methods, and show purity in supplementary information.

We thank the reviewer for this comment. The materials and methods section have been completed, a gel showing the purity of the proteins has been included in the supplementary information.

2. It would be helpful if authors show the chemical structure of ZZW-115 and its possible interaction with NUPR1 in supplementary figure.

We thank this reviewer for this comment; however, this information has been already published in our previous paper (<https://doi.org/10.1172/JCI127223>). Showing its structure here could be redundant.

3. The full name of 5-FU needs to be described in the manuscript for general audience/reader.

Thank you for this suggestion, the text have been modified.

4. There seem to be many typos. Please correct.

Line 151: subtract? Substrate?

Line 205: antioxidant and inhibitor of ferroptosis, Ferrostatin-1,

Line 219: in the presence or not (absence?)

Line 349: Fifth?

We thank for these comments; the text has been corrected.

Reviewer #2 (Remarks to the Author):

PAR (Poly-ADPribose) polymerase 1 (PARP1) activity is tightly regulated and multiples pathways are involved in its regulations in cancer cells. Parthanatos (a regulated cell death, RCD) is mediated by high level of PARP1 activity. In this manuscript, Santofimia-Castaño et al., demonstrated that nuclear protein 1 (NUPR1), a nuclear intrinsically disordered protein (IDP) of 82 amino acids, binds to PAR (Poly-ADPribose) polymerase 1 (PARP1) and inhibits PARP1 activity in pancreatic cancer cells. More importantly, the authors also found that the NUPR1 inhibitor ZZW-115 abolished the protective effect of NUPR1 as a PARP1 activity regulator, inducing strong hyperPARylation and cell death

mediated by a strong mitochondrial dysfunction. This type of cell death is a non-canonical parthanatos, which extends our understanding of NUPR1-regulated biological process in pancreatic cancer as well as the other cancers.

The experiments are well-designed and results are interesting. This reviewer has minor suggestions as follows:

1, The authors claimed that ZZW-115 and PARP1 are competing for the same binding site in NUPR1, involving Ala33 and Thr68. Thus, the mutations of Thr68Gln/Ala33Gln in NUPR1 prevent its inhibition on PARP1 activity as shown in Figure 2B. Does this mean these mutations Thr68Gln/Ala33Gln in NUPR1 lose its binding activity to PARP1? The authors may test their association by proximity ligation assay (PLA) or co-IP or pulldown in vitro.

This is a very interesting point. To demonstrate that NUPR1 is interacting with PARP1 by the same residues as other partners, such as RING1B, or the inhibitors, such as Trifluoperazine or ZZW-115, we have transfected MiaPaCa-2 cells with Flag-NUPR1 (WT, single mutants A33Q or T68Q, or double mutant A33Q/T68Q) and found that single or double mutations decrease significantly the binding between NUPR1 and PARP1 (Supplementary Figure 1B).

2, NUPR1 inhibitor ZZW-115 also induces ferroptosis in a mitochondria-dependent manner, as previously reported in *Cell Death Discov* 2021;7:269. The authors would mention why so many different cell death pathways are connected to NUPR1 inactivation, including senescence in lung cancer and breast cancer (*Cell Death Dis* 2021 Feb 4;12(2):149. *Autophagy*. 2018;14(4):654-70.). Therefore, the authors may put their thoughtful insights of NUPR1 inactivation in Discussion section.

This is a very interesting comment. Our hypothesis is that inhibition of NUPR1 induces multiples irreversible and fatal events that trigger mitochondrial catastrophe, ROS overproduction and loss of ATP levels. These events can trigger multiple cell death pathways concomitantly. We have completed the discussion section.

3, As an IDP and a transcriptional cofactor, NUPR1 may also show its function in gene regulation. It is clear that most of the PLA foci were nuclear location in Figure 1C, the PLA data (mouse anti-Flag and rabbit anti-NUPR1 antibodies). In this case, NUPR1 location may be performed in the presence of ZZW-115 in MiaPaCa-2 cells using rabbit anti-NUPR1 antibody alone, compared to the NUPR1-negative cancer cells.

We thank the reviewer for this comment. Actually, this is a very interesting point that we previously reported (<https://doi.org/10.1172/jci.insight.138117>). ZZW-115 is able to modify the localization of NUPR1 from the nucleus to the cytoplasm by impeding the interaction of NUPR1 with the importins. This effect agrees with the result presented in this manuscript, meaning that by inhibiting the translocation of NUPR1, all the nuclear interactions are hampered.

4. NUPR1 activation could orchestrate a negative feedback with PARP1, limiting the hyperPARylation and its consequent mitochondrial catastrophe dependent cell death, which was reversed by olaparib (PARP1 inhibitor) or the NAD⁺ precursor nicotinamide mononucleotide (NMN). That is ROS-related signaling process, connecting the mitochondrial dysfunction to various cell death pathways. It has been known that NUPR1 is a stress-inducible protein and/or chemotherapy resistance-related protein, the authors might discuss this feature of its dependency such as on germinal BRCA mutations or more widely personal cancer treatment, in the Discussion section.

We fully agree with this comment. We have highlighted the fact that ZZW-115 might be administered as a personalized therapy, but not in patients with BRCA mutation.

Reviewer #3 (Remarks to the Author):

Review summary on the “NUPR1 protects against hyperPARylation-dependent cell death” manuscript

Patricia Santofimia-Castaño and her coworkers investigate the protective role of NUPR1 against hyperPARylation-dependent cell death. Overall, the manuscript is well-written, the majority of the results are convincing. One of the strongest points of the manuscript is that the authors use their formerly developed small-compound (ZZW-115) that inhibits the activity of NUPR1. The authors use 4 cell lines to highlight their results, and the canonical points of the manuscript were confirmed by NUPR1 knockdown. The introduction and the results are clear and concise, and the discussion is articulate. However, I have some concerns about the experiments.

Major comments:

Materials and methods:

In the cell viability section, the authors state that “Medium was discarded”. Due to medium removal, you discard detached dead cells as well, so you will only measure the remaining, theoretically viable cells. However, the crystal violet assay assumes that all cells attached to the plate are viable which is not always true. In this case, Figure 3 does not exclusively represent cell death, instead it may refer to cell cycle arrest as well. PARP1 inhibitors are strong cell cycle arrest inducers. I would suggest to conform this result with another cell death detection method. For instance, using Propidium iodide - Annexin V double labelling followed by flow cytometry would have been a better choice. Or at least try to exclude the possibility of cell cycle arrest involvement by BRDU or Propidium Iodide incorporation.

We thank for this comment. We totally agree with the reviewer that crystal violet might have some disadvantages to measure cell viability. However, our aim was to use a viability method independent of cell metabolism, since ZZW-115 induced mitochondrial dysfunction, thus MTT of Presto Blue assays would also have great disadvantages. We have performed Annexin V/PI staining, that are shown in the Supplementary Figure 3. The results are quite similar with those obtained with the crystal violet assay.

Measurement of OXPHOS. It won't turn out if the authors used normalization after measuring OXPHOS. Normalization in connection with Seahorses studies is a must especially when a treatment induces cell death. The detected lower oxygen consumption in response to ZZW-115 treatment might originate from lower cell viability and OCR values may merely represents cell death as seen by crystal violet assay. I would suggest to do normalization by measuring protein content after performing Seahorse. For example, dissolving the cells in NaOH or lysis buffer and performing a BCA assay. Another option is to present OCR/ECAR values on the graph regardless of normalization. However, it must be noted that OCR/ECAR might somewhat have a different meaning and the results should be interpreted accordingly.

We are sorry for the mistake; levels of OCR and ECAR are always systematically normalized to the cell number, however this information was missing in the previous version, Figures and M&M section have been corrected/completed.

Minor comments.

Introduction:

Line 86: “and consequently inhibits the nuclear translocation and therefore its nuclear activity”. In my opinion the hampered nuclear translocation of a protein does not necessarily result in decreased protein activity. Although, this is true for NUPR1, I would omit „therefore”.

We have omitted “therefore”

Results:

In Figure 2 and 5 I would add separately which statistical tests were used for Figure 5A and 5C and for Figure 2 A/B and Figure 2E. I assume it is not the same as stated in the Figure legend.

We have corrected the text of the figures.

Figure 2D: The name of the housekeeping protein used for normalization is missing below anti-PAR.

We thank for this comment, the protein is β -actin.

Figure 3 might be misleading. If someone is not familiar with crystal violet staining, they won't find out if the graphs represent cell death or cell viability. I would at least mention in the figure legend what does the Y axis represent. I also don't see the increasing concentrations of ZZW-115 written in the figure legend. The same applies for Figure 5.C. It would have also been better in Figure 3 to use the “control” for the non-treated cells, the second group would have been the NUPR1 inhibited cells, the 3. one NUPR1 + olaparib and the 4. one NUPR1 + PDD00017273. Thereby, one could easily compare the extent of cell death among the different groups.

We thank the reviewer for this comment. We have changed the figures and the legends accordingly to the reviewer suggestion

Line 175: “strong disorganization” referring to Figure 4B: Although, the morphological alteration of the mitochondrial network is obvious, rounded-like structures appear in response to ZZW-115 treatment, it would be nice to characterize the morphology of mitochondria. I would suggest to calculate for example mitochondrial circularity, form factor, aspect ratio etc. and add these parameters to the given figure. Images can be analyzed by the ImageJ software Mito-Morphology Macro among others. I would also check some mitochondrial dynamics regulator factors (Mfn1, Mfn2, Mff, Drp1, etc.) either by qPCR or by Western blot. Thereby, you could get an insight into the fission-fusion machinery and confirm the mitochondrial morphology with expressional changes.

We have completed our results by performing quantitative analysis of the morphology of the mitochondria. Results are presented in Supplementary Figure 4.

I suggest to add the NUPR1 siRNA western blot picture either on Figure 6 or in the supplementary file and calculate the knockdown efficiency.

We thank for this comment. A western blot using the same siRNAs and the same cells was already published in the supplementary information of our previous work (<https://doi.org/10.1038/s41598-018-35020-3>). A RT-qPCR 48 hours post transfection is included. The efficient is 88%.

Figure 6B: It is a little confusing how you measured ATP content. Did you calculate ATP level from basal OCR by subtracting OCR values after oligomycin injection or did you perform ATP measurement by an ATP kit? It should be added in the M&M section.

We are sorry for this mistake; we have included in the M&M section the protocol and kit used for this assay.

The graphical abstract is well-made and easy to understand. Do you have any theory how mitochondrial stress leads to ROS production? Does mitochondrial ROS itself induce PARP1 activation? I would be nice to check in Figure 5A how PARP1 activation is affected upon treatment with a specific mitochondrial ROS scavenger (MitoTEMPO) to confirm if ROS with mitochondrial origin triggers PARP1 activation.

We agree with this suggestion and consequently we have performed an experiment with Mito-Q antioxidant. In a previous work, we found that this mitochondrial antioxidant was also able to rescue cell viability upon ZZW-115, since mitochondrial ROS are also highly involved in ZZW-115 dependent cell death. In this line, Mito-Q was also able to decrease PARP1 activity as it is shown in Figure 5A, thus mitochondrial ROS contribute to PARP1 activation.

Materials and methods:

Line 340: Please define the manufacturer of the nitrocellulose membrane.

Line 352: Please define the permeabilization solution.

Line 356, 374 and 397: Please define the version number of ImageJ software.

Line 369 and 378: Please define the manufacturer of ZZW-115, olaparib, 5-FU, PDD00017273 and NMN.

Line 374: Please define the manufacturer of Epoch™ Microplate Spectrophotometer.

Line 387: Please define the manufacturer of Tristar multimode microplate reader.

Line 390: Please define the fixation solution.

Line 403: Please define the manufacturer of accutase solution.

Line 414 and 420: Please define the version number of FlowJo software.

Line 425: Please define the manufacturer of Prolong Gold antifade reagent with DAPI solution.

Line 430: Please define the manufacturer of Seahorse Bioscience XF24 Extracellular Flux Analyzer.

In the statistical analysis part, I would add which test was used to determine the normality of the groups in the case of ANOVA (Kolmogorov-Smirnov test, Shapiro-Wilk test etc.), as it affects if parametric or non-parametric test must be used. Please define the program and its version number used for statistical evaluation.

Seeded cell density are missing in some paragraphs in the methods section.

The meaning of the abbreviations sometimes is not obvious and they should be mentioned for example in the case of DTAB, 5-FU, HBSS, OXPHOS etc.

We thank this reviewer for these corrections, all the information requested have been included

Evaluation:

Patricia Santofimia-Castaño and her coworkers' manuscript provides valuable data on the NUPR1 inactivation- and PARP1-mediated cell death involving mitochondrial dysfunction. The results embrace diverse methodological approach and they lay the foundation for further studies. However, I would like to ask the authors to address my concerns and comments.

Reviewers' comments:

Reviewer #1 (Remarks to the Author):

This reviewer recommends two experiments before accepting this manuscript.

1. Regarding the major point 1:

Both CD and fluorescence do not show significant changes. The reduction in CD in lower wavelength can be attributed to other changes than structural changes, which should accompany with significant change in shapes of curves, etc.

Since authors have recombinant proteins for NUPR1 and PARP1, the easiest way is to do in vitro IP using recombinant proteins to see there are any direct interactions in purified proteins. Related to major point 2, I strongly recommend to test interactions between NUPR1 and unmodified & PARylated PARP1 protein.

2. Regarding the major point 2:

Cells have basal level of PARylation, and treating Olaparib to FU-untreated cell would show much less effects.

Authors should run an experiment comparing PLA for FU, Olaparib, and FU + Olaparib. These cells can be also monitored with anti-PAR antibody to correlate with the level of PAR in cells for each sample.

I believe This experiment will address the PAR-dependency of this interaction.

Other points are addressed in the revised manuscript.

Reviewer #2 (Remarks to the Author):

Having looked at the revised manuscript, I believe the authors have done a very good job at providing a more solid and more coherent paper. I am now happy to support publication. I think this will be an important addition to the field of NUPR1 inhibition in cancers.

Reviewer #3 (Remarks to the Author):

Review summary on the 'NUPR1 protects against hyperPARylation-dependent cell death' manuscript

Patricia Santofimia-Castaño and her coworkers have made the revised version of their manuscript according to my review summary. The authors put great effort into amending the mistakes and their results were confirmed with additional figures. Since all my questions and concerns has been addressed, I recommend to accept the manuscript for publication.

Reviewers' comments:

Reviewer #1 (Remarks to the Author):

This reviewer recommends two experiments before accepting this manuscript.

1. Regarding the major point 1:

Both CD and fluorescence do not show significant changes. The reduction in CD in lower wavelength can be attributed to other changes than structural changes, which should accompany with significant change in shapes of curves, etc. Since authors have recombinant proteins for NUPR1 and PARP1, the easiest way is to do in vitro IP using recombinant proteins to see there are any direct interactions in purified proteins. Related to major point 2, I strongly recommend to test interactions between NUPR1 and unmodified & PARylated PARP1 protein.

We agree with the reviewer comment and therefore we have performed the Co-IP with the recombinant proteins. In this experiment, we have been able to determine that NUPR1 and PARP1 directly interact in vitro; this new data is included in the Supplementary Figure 1C.

Interactions of NUPR1 with PARP1 unmodified or PARylated have been addressed in cells, since recombinant PARylated PARP1 was not available, by the PLA assays, evaluating the interaction between NUPR1 and PARP1 (Figure 1C) and between NUPR1 and PAR (Supplementary Figure 1A).

2. Regarding the major point 2:

Cells have basal level of PARylation, and treating Olaparib to FU-untreated cell would show much less effects. Authors should run an experiment comparing PLA for FU, Olaparib, and FU + olaparib. These cells can be also monitored with anti-PAR antibody to correlate with the level of PAR in cells for each sample. I believe This experiment will address the PAR-dependency of this interaction.

We found this comment very relevant. We have included the data of the NUPR1/PARP1 interaction by PLA assay of cell treated by 5FU+Olaparib. In our experiments, we observed that even under PARP1 inhibition by Olaparib, there is a significant interaction between both proteins (alone or in combination with 5-FU). Moreover, either by immunofluorescence experiment or by Western Blotting, we have monitored the PAR levels, which are non-detectable by any of these assays, indicating that the interaction between PARP1 and NUPR1 is independent of the PARylated state of PARP1.

Other points are addressed in the revised manuscript.

We thank to this reviewer for her/his positive comment.

Reviewer #2 (Remarks to the Author):

Having looked at the revised manuscript, I believe the authors have done a very good job at providing a more solid and more coherent paper. I am now happy to support publication. I think this will be an important addition to the field of NUPR1 inhibition in cancers.

We also thank to this reviewer for her/his positive comment..

Reviewer #3 (Remarks to the Author):

Review summary on the 'NUPR1 protects against hyperPARylation-dependent cell death' manuscript

Patricia Santofimia-Castaño and her coworkers have made the revised version of their manuscript according to my review summary. The authors put great effort into amending the mistakes and their results were confirmed with additional figures. Since all my questions and concerns has been addressed, I recommend to accept the manuscript for publication.

We thank to this reviewer for her/his positive comment.

REVIEWERS' COMMENTS:

Reviewer #1 (Remarks to the Author):

Authors successfully addressed all this reviewer's concerns and suggestions. I am happy to recommend to accept this manuscript.